# NEFTune: Noisy Embeddings Improve Instruction Finetuning

**Neel Jain**[1*], **Ping-yeh Chiang**[1*], **Yuxin Wen**[1*], **John Kirchenbauer**[1], **Hong-Min Chu**[1],
**Gowthami Somepalli**[1] , **Brian R. Bartoldson**[2], **Bhavya Kailkhura**[2], **Avi Schwarzschild**[1],
**Aniruddha Saha**[1], **Micah Goldblum**[3], **Jonas Geiping**[1], **Tom Goldstein**[1]
[1] University of Maryland, [2] Lawrence Livermore National Laboratory, [3] New York University

## Abstract

We show that language model finetuning can be improved, sometimes dramatically, with a simple augmentation. `NEFTune` adds noise to the embedding vectors during training. Standard finetuning of LLaMA-2-7B using Alpaca achieves 29.79% on AlpacaEval, which rises to 64.69% using noisy embeddings. `NEFTune` also improves over strong baselines on modern instruction datasets. Models trained with Evol-Instruct see a 10% improvement, with ShareGPT an 8% improvement, and with OpenPlatypus an 8% improvement. Even powerful models further refined with RLHF such as LLaMA-2-Chat benefit from additional training with `NEFTune`. Particularly, we see these improvements on the conversational abilities of the instruction model and not on traditional tasks like those on the OpenLLM Leaderboard, where performance is the same.

## 1 Introduction

Generative language models are typically trained on raw web data, and then subsequently fine-tuned on a comparatively small but carefully curated set of instruction data. Instruction fine-tuning is crucial to taming the power of LLMs, and the usefulness of a model is largely determined by our ability to get the most out of small instruction datasets.

In this paper, we propose to add random noise to the embedding vectors of the training data during the forward pass of fine-tuning. We show that this simple trick can improve the outcome of instruction fine-tuning, often by a large margin, with no additional compute or data overhead. Noisy Embedding Instruction Fine Tuning (`NEFTune`), while simple, has a strong impact on downstream conversational quality. When a raw LLM like LLaMA-2-7B is finetuned with noisy embeddings, its performance on `AlpacaEval` improves from 29.8% to 64.7% (Figure 1) – an impressive boost of around 35 percentage points (Touvron et al., 2023b; Dubois et al., 2023). `NEFTune` leads to this surprising and large jump in performance on conversational tasks, maintaining performance on factual question answering baselines.

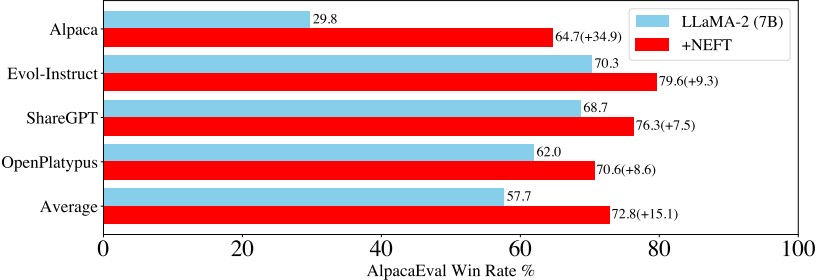

Figure 1: Alpaca Win Rate measurements for LLaMA-2-7B models finetuned on various datasets with and without `NEFTune`. `NEFTune` leads to performance boosts across all of these datasets, showcasing the increased conversational quality of the generated answers.

---

\* Equal contribution. Code is available on Github: `https://github.com/neelsjain/NEFTune`.
Correspondence to Neel Jain: <njain17@umd.edu>.

## 1.1 RELATED WORK

The earliest forms of instruction finetuning such as FLAN and T0 (Sanh et al., 2021; Wei et al., 2021) focused on cross-task generalization in language models. Encoder-decoder language models were finetuned on a broad range of NLP tasks (about 100) and then evaluated on a set of different tasks. This was later scaled up to include thousands of tasks, seeing further improvement over the original FLAN (Chung et al., 2022; Xu et al., 2022). Although these works showed that LLMs could be easily adapted to solve simple and classical NLP tasks, real-word scenarios require LLMs to provide free-form answers to open-ended queries.

InstructGPT (Ouyang et al., 2022) was the first model to tackle open-ended queries with impressive performance. OpenAI further trained GPT-3 (Brown et al., 2020) using reinforcement learning from human feedback (RLHF) to *align* the model. This procedure gave rise to highly popular models like ChatGPT (OpenAI, 2022) that captured the imagination of the general public and generated longer coherent text than its InstructGPT predecessor.

This led to the work of Wang et al. (2022) (*Self-Instruct*), which used InstructGPT (Text-Davinci-003) to produce instruction-output pairs which could be used to finetune the foundation models like LLaMA into instruction following variants like Alpaca (Taori et al., 2023). Through the rise in popularity of distilled models Taori et al. (2023), the community has constructed other datasets distilling in particular ways from other models like ChatGPT, including Xu et al. (2023). In another approach, ShareGPT (Chiang et al., 2023) was constructed by crowd sourcing real user conversations from ChatGPT. Other datasets like Lee et al. (2023) construct a dataset to improve specific aspects of the model like STEM question answering and logical reasoning. AlpaGasus (Chen et al., 2023) filters data by quality (according to GPT-4) to improve performance.

It should be noted that noisy inputs have been used to improve models in various ways. The first instance of noise being used to improve language models was the FreeLB method by Zhu et al. (2019), who observed that adversarial perturbations boosted the performance of MLM models. The noise in this case is not random, but is rather computed by first adding a small Gaussian perturbation to the embeddings and then using a gradient step to find the perturbation that maximally alters model performance. This adversarial augmentation approach also improves model performance on graphs Kong et al. (2022). While our proposed scheme is non-adversarial, we adopt the noise scaling rules from these works. Training on noisy inputs has also been done for other applications, such as to improve image captioning systems (Nukrai et al., 2022; Roth et al., 2023), natural language understanding tasks (Wu et al., 2022; Yuan et al., 2023; Roth et al., 2023), and as a common component of early differential privacy mechanisms (Dwork et al., 2014).

## 2 NEFTUNE: NOISY EMBEDDING INSTRUCTION FINETUNING

Instruction models are trained on datasets comprising pairs of instructions and responses. Each step of NEFTune begins by sampling an instruction from the dataset, and converting its tokens to embedding vectors. NEFTune then departs from standard training by adding a random noise vector to the embeddings. The noise is generated by sampling iid uniform entries, each in the range $[-1, 1]$, and then scaling the entire noise vector by a factor of $\alpha/\sqrt{Ld}$, where $L$ is the sequence length, $d$ is the embedding dimension, and $\alpha$ is a tunable parameter. This scaling rule was borrowed from the adversarial ML literature (Zhu et al., 2019; Kong et al., 2022), and results in a random vector with an expected Euclidean magnitude of approximately $\alpha/\sqrt{3}$. Algorithm 1 describes our method in detail.

## 3 EXPERIMENTAL SET-UP

### 3.1 MODELS

We conduct the majority of our experiments using 7B parameter LLMs. Particularly, we use LLaMA-1, LLaMA-2, and OPT-6.7B (Touvron et al., 2023a;b; Zhang et al., 2022). These similarly shaped transformers mostly differ in tokens seen during training. OPT, LLaMA-1, and LLaMA-2 were trained using 180B, 1T, and 2T tokens respectively. This difference is to be reflected in model performance on standard benchmarks like MMLU, with LLaMA-2 performing the best and OPT performing the worst. For the 13B parameter models, we train LLaMA-2. Additionally, we improve RLHF models by finetuning the highly refined LLaMA-2-Chat (7B) model.

---

**Algorithm 1** NEFTune: **N**oisy **E**mbedding Instruction **F**ine**tun**ing

---

**Input:** $\mathcal{D} = \{x_i, y_i\}_1^N$ tokenized dataset, embedding layer $\text{emb}(\cdot)$, rest of model $f_{/\text{emb}}(\cdot)$, model parameters $\theta$, $\text{loss}(\cdot)$, optimizer $\text{opt}(\cdot)$
NEFT Hyperparameter: base noise scale $\alpha \in \mathbb{R}^+$

Initialize $\theta$ from a pretrained model.

**repeat** $(X_i, Y_i) \sim \mathcal{D}$          ▷ sample a minibatch of data and labels
     $X_{\text{emb}} \leftarrow \text{emb}(X_i), \mathbb{R}^{B \times L \times d}$      ▷ batch size $B$, seq. length $L$, embedding dimension $d$
     $\epsilon \sim \text{Uniform}(-1, 1), \mathbb{R}^{B \times L \times d}$      ▷ sample a noise vector
     $X'_{\text{emb}} \leftarrow X_{\text{emb}} + (\frac{\alpha}{\sqrt{Ld}})\epsilon$      ▷ add scaled noise to embeds [a]
     $\hat{Y}_i \leftarrow f_{/\text{emb}}(X'_{\text{emb}})$      ▷ make prediction at noised embeddings
     $\theta \leftarrow \text{opt}(\theta, \text{loss}(\hat{Y}_i, Y_i))$      ▷ train step, e.g., grad descent
**until** Stopping criteria met/max iterations.

---

[a]If sequence lengths in a batch are not equivalent, then $L$ is a vector $\in \mathbb{Z}_{>0}^B$ and the scaling factor $(\alpha/\sqrt{Ld})$ is computed independently for each sequence in batch.

---

## 3.2 Instruction Finetuning Datasets

We focus on the following finetuning datasets either because of their wide popularity, or because they have yielded state-of-the-art results in the recent past. Note that we use only single-turn datasets because of the memory constraints of our hardware setup.

- **Alpaca** (Taori et al., 2023) was constructed using the *Self-Instruct* method of Wang et al. (2022), and the Text-Davinci-003 model (Ouyang et al., 2022). Self-Instruct uses a small seed set of tasks to construct new instruction tuning tasks and filter out bad ones.

- **Evol-Instruct** (Xu et al., 2023) contains 70k single-turn instructions that are considered more complex than Alpaca. This dataset was derived from the Alpaca dataset by using ChatGPT to *evolve* the initial instructions.

- **Open-Platypus** (Lee et al., 2023) is a curated dataset amalgamated from 11 open-source datasets, curated specifically towards improving LLM performance in STEM and logical domains. This set contains 25k questions where $\approx 10\%$ are LLM-generated and the remainder human-written.

- **ShareGPT** (Chiang et al., 2023) is a dataset of 70K voluntarily-shared ChatGPT conversations (ShareGPT, 2023). Although ShareGPT is multiturn, we use the dataset version from Vicuna-v1.1 and split the multi-turn conversations closer to a single-turn format.

Additionally, we finetune all models with the Alpaca system prompt, except for ShareGPT, where we use the Vicuna system prompt. The hyperparameters can be found in Appendix A.1. We set our hyperparameters through a coarse sweep on LLaMA-1 (7B) trained on the Alpaca dataset, where we see 6% improvement over the standard Alpaca model. We use these as the defaults on all models.

## 3.3 Evaluation

Since we train using largely single-turn data, we evaluate the model's conversational abilities using AlpacaEval. We also evaluate on the tasks from the OpenLLM Leaderboard to determine if the NEFTune augmentation causes any loss in performance on standard multiple choice tasks.

**AlpacaEval.** The AlpacaEval dataset released by Dubois et al. (2023) is used to evaluate the overall quality of generations. AlpacaEval is an automatic model-based evaluation that compares Text-Davinci-003 generations to the model generations over 805 instructions with the Win Rate reported. The Win Rate is the rate at which the model in question is preferred to Text-Davinci-003 as determined by model evaluator (GPT-4). The 805 test prompts are scraped from vicuna, koala, Anthropic's hh-rlhf, and other sources, making it a fairly comprehensive and diverse test. Additionally, AlpacaEval has high agreement with humans (Dubois et al., 2023) (validated on 20K annotations). We believe at the 7B and 13B scale this evaluation is still quite reasonable. We use both GPT-4 and ChatGPT as evaluators. We use ChatGPT as a precursor test to determine which models to evaluate on GPT-4. This is due to the cost and API restrictions of GPT-4.

Table 1: `AlpacaEval` Win Rate versus Text-Davinci-003 for LLaMA-2 trained on different datasets, using GPT-4 as the evaluator, showing an average improvement of 15% across all datasets over three seeds.

|  | Alpaca | Evol-Instruct | ShareGPT | OpenPlatypus | Average |
|---|---|---|---|---|---|
| Llama-2 7B | $29.91_{\pm0.40}$ | $70.19_{\pm0.08}$ | $70.57_{\pm1.38}$ | $60.95_{\pm1.52}$ | 57.91 |
| +`NEFTune` | $65.70_{\pm0.52}$ | $80.36_{\pm0.47}$ | $76.22_{\pm0.67}$ | $69.96_{\pm0.47}$ | 73.06 |

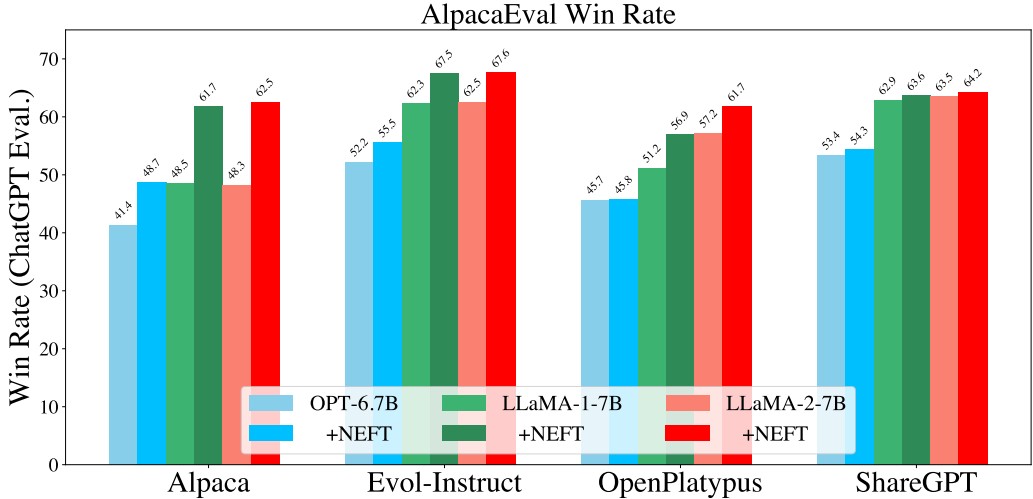

Figure 2: `AlpacaEval` Win Rate with and without `NEFTune` on LLaMA-2, LLaMA-1, and OPT across Alpaca, Evol-Instruct, ShareGPT and OpenPlatypus datasets. Performance improves across different datasets and models with ChatGPT as the evaluator.

**Hugging Face OpenLLM Leaderboard.** The evaluation datasets used for leaderboard ranking are the verbalized multiclass classification datasets ARC (Clark et al., 2018), HellaSwag (Zellers et al., 2019), MMLU (Hendrycks et al., 2020), and TruthfulQA (Lin et al., 2022). This combination of datasets broadly evaluates the ability of a LLM to respond to factual questions and reasoning challenges, and we evaluate on these datasets to confirm that model capabilities are not negatively impacted by `NEFTune`.

## 4 RESULTS

**`NEFTune` Improves Text Quality.** From Table 1, we can see an increase across all datasets for the 7B scale with an average increase of 15.1%, showing that training with `NEFT` significantly improves conversational ability and answer quality, as measured via `AlpacaEval`. Additionally, we can see from Figure 2 that we also see improvements on older models, such as LLaMA-1 and OPT. Interestingly, we see less improvement on ShareGPT than on other datasets according to ChatGPT. However, this is not reflected in the GPT-4 evaluation. From Table 2, we see the Win Rate climbs from 75.03% to 88.81% (+13.78%) after adding `NEFTune` to the 70B parameter model trained on Evol-Instruct (hyperparameters in Appendix A.1).

**`NEFTune` Can Improve Chat Models.** From Table 2, we see that further instruction finetuning LLaMA-2 Chat (7B) on Evol-Instruct can boost the performance of LLaMA-2-Chat by 3%. This model was already extensively tuned, using multiple rounds of RLHF. Yet, with `NEFTune`, we see a sizable, additional performance increase of 10%, although we note that some capabilities of this checkpoint model may be affected like its ability to refrain from outputting toxic behavior. Nevertheless, it is surprising that the conversation quality of such a refined chat model can be so dramatically improved.

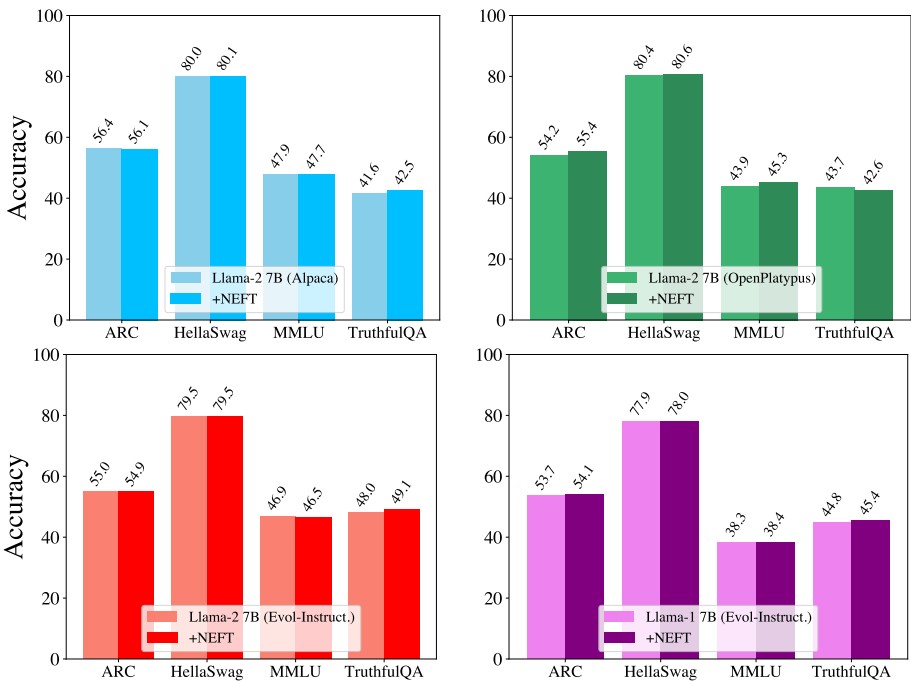

Figure 3: OpenLLM Leaderboard tasks with and without `NEFTune` on LLaMA-2 across Alpaca, Evol-Instruct, and OpenPlatypus datasets and LLaMA-1 trained on Evol-Instruct. We observe that performance does not change across datasets and models.

**Effect on Capabilities.** A potential concern is that `NEFTune` improves conversational ability only at the cost of other classical skills. We evaluate on the *OpenLLM Leaderboard* tasks, using the LM-Eval Harness (Gao et al., 2021) implementation of MMLU, ARC, HellaSwag, and TruthfulQA. These benchmarks give us a glimpse into model knowledge, reasoning, and truthfulness. Figure 3 shows that scores remain stable and that `NEFTune` preserves model capabilities.

Table 2: LLaMA-2-Chat (7B), LLaMA-2 (13B), and LLaMA-2 (70B) can be finetuned further to improve performance.

|  | LLaMA-2 (7B) | LLaMA-2-Chat (7B) | LLaMA-2 (13B) | LLaMA-2 (70B) |
|---|---|---|---|---|
| Base | - | 71.37* | - | - |
| Evol-Instruct | 70.34 | 74.44 | 72.61 | 75.03 |
| +`NEFT` | 79.60 | 81.74 | 82.04 | 88.81 |

**`NEFTune` Works with QLORA.** We show that `NEFTune` also improves performance in constrained resource environments by training with Quantized Low Rank Adapters (QLORA) (Dettmers et al., 2023). We use the implementation from Dettmers et al. (2023), and the default training hyperparameters for all model weights, training for only one epoch. Table 3 shows that when training with QLORA, `AlpacaEval` performance increases across all model sizes and datasets studied. However, performance gains are less stark than those seen in full scale finetuning. This may be because different hyperparameters (i.e, number of finetuning epochs) are needed, or because we are heavily quantizing to 4-bits.

**A Qualitative Example.** Here we show a qualitative example from LLaMA-2 trained on Alpaca with and without `NEFT`. We choose the *quantum computing prompt* from the Vicuna Eval subset of the `AlpacaEval` instructions. We see from this example that Alpaca produces a response that is shorter, giving only a very basic definition of quantum computing, mentioning qubits, superposition, entanglement, and increased complex computation. In the Alpaca-`NEFT` response, the model

Table 3: `AlpacaEval` Win Rate (ChatGPT Eval.) reported across different datasets and model sizes. Even training with QLORA, we can see performance increases across the board, although they are milder than regular finetuning.

| Model
Dataset | LLaMA2(7B)
Alpaca | LLaMA2(7B)
Evolve70k | LLaMA2(13B)
Alpaca | LLaMA2(13B)
Evolve70k | LLaMA1(30B)
Alpaca |
|---|---|---|---|---|---|
| Base | 26.71 | 52.80 | 31.99 | 56.96 | 31.99 |
| +`NEFT` 5 | 31.43 | 56.96 | 38.20 | 60.99 | 35.16 |
| +`NEFT` 10 | 28.57 | 54.84 | 41.55 | 61.30 | 36.89 |
| +`NEFT` 15 | 29.01 | 57.27 | 42.17 | 57.39 | 35.03 |

provides a more fluid answer, expounding on these topics with clearer explanations of superposition and quantum entanglement, and including a mention of potential applications. We feel this example is representative of the kinds of changes induced by `NEFT`. See the appendix for more examples using different finetuning datasets.

> **`AlpacaEval` (Vicuna Subset) Prompt**: Can you explain the basics of quantum computing?
>
> **Alpaca-2-7B Response**:
> Quantum computing is based on the principles of quantum mechanics, which allow for phenomena such as superposition and entanglement between particles. This means that information can be processed in parallel instead of sequentially like traditional computers. Quantum bits (qubits) are used to store data instead of classical bits, allowing more complex computations to take place with fewer resources than would otherwise be necessary.
>
> **Alpaca-2-7B-`NEFT` Response**:
> Quantum computing is based on the principles of quantum mechanics, which describe how particles such as electrons and photons behave at the subatomic level. Unlike traditional computers, which process information in binary digits (bits) that can be either 0 or 1, quantum bits (qubits) can exist in multiple states between 0 and 1 simultaneously. This phenomenon is known as superposition and allows for much more complex calculations to be performed than would otherwise be possible with classical computers. Quantum algorithms are designed to take advantage of this property by encoding data into qubit states so that it can be processed exponentially faster than conventional methods. Additionally, quantum entanglement–the ability for two particles to remain connected even over great distances–allows for large amounts of data to be sent across networks quickly and securely. The potential applications of quantum computing range from cryptography and artificial intelligence to drug discovery and material science. With continued research and development, quantum technology has the potential to revolutionize many industries.

# 5 ANALYSIS

We hypothesize that by adding noise to the embeddings at train time, the model overfits less to the specifics of the instruction-tuning dataset, such as formatting details, exact wording, and text length. Instead of collapsing to the exact instruction distribution, the model is more capable of providing answers that incorporate knowledge and behaviors of the pretrained base model.

A very noticeable side-effect of this, that we observe immediately, is that the model is forming more coherent, longer completions. Longer, more verbose, completions are preferred by both human and machine evaluators on most datasets (Dubois et al., 2023), but we find that the increased verbosity is only the most visible side-effect from the reduced overfitting to the instruction distribution; increased verbosity alone cannot explain the measured gains in performance.

## 5.1 OVERFITTING

In this analysis, we focus on LLaMA-2-7B models trained on the Alpaca dataset both with and without `NEFTune`. We examine the training loss of both models on the Alpaca dataset (both are

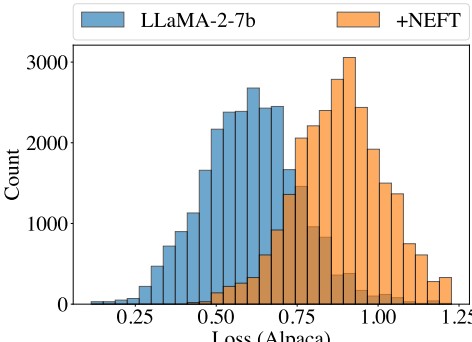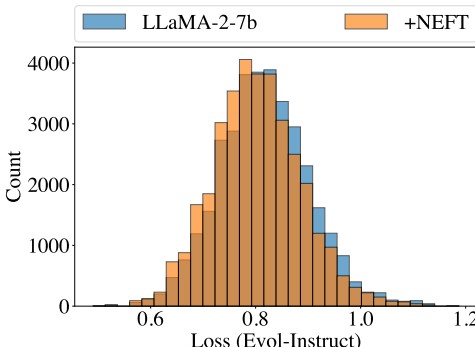

Figure 4: **Left:** training loss on the *Alpaca dataset* for models with and without NEFT, computed with no added noise. Training with NEFT yields a higher training loss. **Right:** loss of the same model, but evaluated on the "test" Evol-Instruct dataset. NEFT yields slightly lower loss.

evaluated without noise) and the "testing" loss on the Evol-Instruct dataset. See Figure 4, which shows that the NEFTune model has significantly higher training loss but slightly lower testing loss compared to the base model trained without NEFTune. This indicated less overfitting and better generalization when NEFTune is used.

To test our overfitting hypothesis further, we also generate responses to training prompts with these models using greedy decoding. We compare the generated responses with the ground truth responses provided in the dataset and report the results in Figure 5. We use ROUGE-L (Lin, 2004) and BLEU (up to n-gram order 4) (Papineni et al., 2002) to measure the similarity between responses. Figure 5 shows that responses by the model trained with NEFTune have significantly lower ROUGE-L and BLEU scores. As ROUGE-L is based on longest common subsequence of words and BLEU is based on common n-grams between responses, higher scores on responses generated by the model trained without NEFT indicate that its responses contain a significantly larger portion of the same words in the same order from the ground truth response, as compared to the outputs of the model trained without NEFTune.

Taken together, these observations imply that standard finetuning recipes, while tuned for maximal performance, significantly overfit to the instruction dataset, inducing exact reproduction of some responses. In contrast, NEFTune models overfit less without reduction in performance on the test set, and do not "lock-in" to the exact wording of the instruction data, as seen in the ROUGE-L metric.

## 5.2 LENGTH VERSUS TOKEN DIVERSITY

Due to the strong correlation between increased length and performance on the AlpacaEval task (in our experiments and for submissions to the public leaderboard), we wondered whether the increase in length observed with NEFTune might come at a cost to the diversity of the text. To investigate this, we compute the n-gram repetition rates for LLaMA-2 trained on different finetuning datasets with and without NEFT[1]. N-grams reoccur more frequently in longer passages, and so we must control for passage length. We compute repetition and diversity scores on a fixed-length chunk at the beginning of each sample. The fixed length cutoffs were 50 for models trained on Alpaca, 100 for Evol-Instruct, 150 for ShareGPT, and 150 for OpenPlatypus. We choose the chunk lengths so that at least half of the generations were longer than the cuttoff, and sequences of insufficient length were dropped. The diversity scores we compute are a summary measure of 2-, 3-, and 4-gram repetition rates called *log-diversity*, as described in Kirchenbauer et al. (2023); Li et al. (2022).

In Table 4 and Table 6, we see that NEFT models generate longer outputs than their counterparts. However, we also see that the 2-gram repetition rates as well as overall token log-diversity for models

---

[1]Note that for all models we performed generation with a repetition penalty of 1.2, held constant across all experiments.

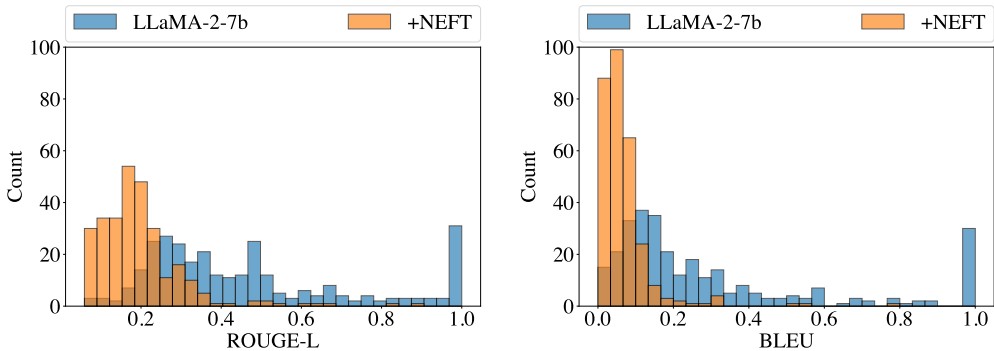

Figure 5: **Left** shows the ROUGE-L of training with and without `NEFT`. **Right** shows BLEU score.

trained with and without `NEFT` are nearly identical, providing evidence that the longer responses do not come at the expense of repetition, and instead provide additional details.

### 5.3 LENGTH IS (NOT) ALL YOU NEED

To scrutinize the length–leaderboard correlation even further, we tested whether simply promoting a model to generate longer outputs was sufficient to recover the performance gains of models trained with `NEFT`. See Table 5. First, we try explicitly prompting the model to give longer answers. Interestingly, this boosts AlpaceEval scores by 16%. We can also coerce long completions by blocking the [EOS] token until we hit 250 tokens in length, thus forcing a standard model to produce answers as long as `NEFT`. This results in marginal improvements over standard finetuning.

Finally, we ablate the use of uniform versus Gaussian noise in the `NEFT` algorithm and find that Gaussian noise induces even longer outputs, but does not come with improved performance. See Table 6. While longer generations do score better, we see that no prompting strategy that we tested came close to the performance of `NEFTune` models.

Table 4: (**Row 1**) Avg. Character lengths of `AlpacaEval` responses from LLaMA-2 models fine-tuned on different datasets. We also report average output length for each dataset (though we trained with max sequence length of 512). `NEFT` increases average length. (**Row 2**) Whitespace-tokenized lengths of generations. (**Row 3**) 2-Gram repetition rates. (**Row 4**) Log-Diversity measures.

| | | Alpaca ($\alpha = 5$) | Evol-Instruct ($\alpha = 5$) | ShareGPT ($\alpha = 10$) | OpenPlatypus ($\alpha = 15$) |
|---|---|---|---|---|---|
| **Character Lengths** | Training data | 270.31 | 1356.43 | 1276.76 | 649.39 |
| | LLaMA-2 7B | 375.22 | 864.06 | 1011.28 | 1100.98 |
| | +NEFT | 1061.89 | 1403.59 | 1496.86 | 1694.26 |
| **Whitespace Lengths** | LLaMA-2 7B | 60.5 | 138.99 | 161.04 | 170.41 |
| | +NEFT | 169.36 | 225.56 | 234.99 | 264.12 |
| **2-Gram Repetition %** | LLaMA-2 7B | 1.49 | 3.87 | 4.82 | 2.73 |
| | +NEFT | 1.72 | 3.79 | 4.58 | 3.21 |
| **Log-Diversity** | LLaMA-2 7B | 15.97 | 10.65 | 8.40 | 9.96 |
| | +NEFT | 16.41 | 10.77 | 8.60 | 9.64 |

### 5.4 HUMAN STUDY

Since our primary results are based on the `AlpacaEval` benchmark, which is scored by a large language model, we also run a small scale human study amongst the authors of this work. For a subsample of 140 instructions from `AlpacaEval`, we present annotators with one response generated by a LLaMA-2 model finetuned on Alpaca data with `NEFT` and another response from a model trained without `NEFT`, in random order.

Table 5: We use the following meta-prompts to get longer responses: "Generate a long response", "Generate a comprehensive response", and "Generate a long and comprehensive response." Longer responses score better, but do not close the gap with `NEFT`.

| Setting (LLaMA-1) | GPT-4 Win Rate | Avg. Character Length |
|---|---|---|
| Alpaca-7B-`NEFT` | 61.99 | 1058.46 |
| Alpaca-7B (Long + Comp) | 48.01 | 620.74 |
| Alpaca-7B (Long) | 44.84 | 614.21 |
| Alpaca-7B (Comprehesive) | 42.14 | 494.85 |
| Alpaca-7B (Min New Tokens) | 38.58 | 1110.97 |
| Alpaca-7B | 32.36 | 375.22 |

Table 6: Win Rate (and Avg. Character Length) on `AlpacaEval` as evaluated by ChatGPT for different levels and types of training noise. While length does increase with noise, it is not always indicative of `AlpacaEval` Win Rate.

| Setting | Alpaca | | Evol-Instruct | | OpenPlatypus | |
|---|---|---|---|---|---|---|
| LLaMA-2-7b | 48.26 | (375.22) | 62.55 | (864.06) | 57.20 | (1100.98) |
| +Uniform Noise 5 | 62.55 | (1061.89) | 67.58 | (1403.59) | 60.99 | (1428.31) |
| +Uniform Noise 10 | 61.18 | (1009.94) | 65.59 | (1696.88) | 60.62 | (1833.85) |
| +Uniform Noise 15 | 61.86 | (819.61) | 66.58 | (1650.65) | 61.74 | (1694.26) |
| +Gaussian Noise 5 | 60.93 | (1371.32) | 65.09 | (2065.75) | 59.13 | (2060.92) |

Human annotators preferred `NEFT` in 88 instances, and 22 instances were a draw. This corresponds to a 74.6% win score for `NEFT` using the `AlpacaEval` formula $(88/(140 - 22))$. Next, we performed a modified run of `AlpacaEval` where, instead of asking the evaluator (GPT-4) to choose between the outputs of our model or Text-Davinci-003, we present a pair of responses from the standard finetuned model and a `NEFT` version of the same model. There, we observe a win score of 92.80%.

## 6 CONCLUSIONS AND LIMITATIONS

The success of `NEFTune` points to the often ignored importance of algorithms and regularizers for LLM training. Unlike the computer vision community, which has studied regularization and overfitting for years, the LLM community tends to use standardized training loops that are designed for optimizer stability and not generalization. In this environment, LLM researchers have become fixated on datasets and model scaling as the primary path forward. Given the consistent gains of `NEFTune`, techniques such as regularization and robust optimization may deserve to be revisted in the LLM setting.

Our study has several limitations. We adopt `AlpacaEval` as our central measure of instruction-following ability for LLMs, which is subject to the biases of a single judge (GPT-4). Additionally, due to limited compute resources, we were not able to validate the success of `NEFTune` on larger 30B and 70B variants across different datasets of LLaMA-2, and we had to rely on fixed hyperparameters for most `NEFTune` runs rather than sweeping. Finally, despite our empirical studies, we do not have a conclusive understanding of why `NEFTune` works.

## 7 ETHICS STATEMENT

In this work, we proposed an augmentation for instruction finetuning. Although we evaluate these models on standard benchmarks, we do not rigiously evaluate the impact of `NEFTune` on model safety and reliability characteristics like toxicity or refusal to provide potentially harmful responses.

## 8 REPRODUCIBILITY STATEMENT

We describe the models (in Section 3.1) and datasets (in Section 3.2) used in our experiments including all hyperparameters (in Section A.1). The compute infrastructure used was based on commodity-level CPUs and GPUs running open source software. Additionally, with submission of the review copy of the work we have included a zip archive containing an anonymized copy of the source code developed through the course of the research. We have added a github link to the codebase as well. The codebase is designed to be both useable and extensible for further research and upon publication, a link to a public copy of the source code will be added to the work.

### ACKNOWLEDGMENTS

This work was made possible by the ONR MURI program, the Office of Naval Research (N000142112557), and the AFOSR MURI program. Commercial support was provided by Capital One Bank, the Amazon Research Award program, and Open Philanthropy. Further support was provided by the National Science Foundation (IIS-2212182), and by the NSF TRAILS Institute (2229885).

Furthermore, this work was performed under the auspices of the U.S. Department of Energy by the Lawrence Livermore National Laboratory under Contract No. DE-AC52-07NA27344. Brian Bartoldson's and Bhavya Kailkhura's efforts were supported by the LLNL-LDRD Program under Project No. 24-ERD-010 (LLNL-CONF-855498). We are also grateful to Amar Saini who provided HPC support.

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

# A  APPENDIX

## A.1  HYPERPARAMETERS

We finetune the 7B parameter models on four A5000s and 13B parameters on eight A5000s using bfloat16 precision. After doing an initial learning rate sweep on LLaMA-1 and Alpaca, we use learning rate of $5e$-5 and the Adam optimizer for all 7B models after seeing $4\%$ improvement over baseline numbers. We train all models for 3 epochs on all datasets setting the same seed for each run with an effective batch size of 128 (4 cards, batch size 4, 8 gradient accumulation steps). When finetuning with noise we train on three levels of noise, an L2 ball of 5, 10, and 15 over the sequence lengths and report the best one on `AlpacaEval` using ChatGPT as the evaluator. We train with sequence lengths of 512 tokens (mainly for memory and speed) like the original Alpaca setting, finding that this does not effect the outputted response length or quality as corroborated by the Alpaca Leaderboard numbers. In Table 10 we see that training with longer sequences does not change performance significantly. For ShareGPT, we split the multi-turn conversations into sequences of 512 tokens using the process described by Chiang et al. (2023). When training 70 billion parameter models, we use the finetuning hyperparameters found in Touvron et al. (2023b) except we use a sequence length of 2048; i.e., we use weight decay of 0.1, a batch size of 64, and a learning rate of $2e-5$. We finetune for a total of three epochs on Evol-Instruct 70k (Xu et al., 2023). When using `NEFTune` on the 70B parameter model, we use $\alpha = 15$ and did not explore other (potentially better) settings due to computational constraints. Additiononally, we saw an increase in average output character length from 852 to 1241 (+389).

## A.2  ADDITIONAL ABLATION STUDIES

We ablated uniform and guassian noise, finding that uniform noise performs slightly better. We also ablate the decoding hyperparameters in Figure 7 finding minimal changes in performance. Thus, we use the simpliest sampling strategy, greedy decoding, with a repetition penalty of 1.2. We also check to see if `NEFT` continues to yield improvments as you increase the number of training epochs. From Table 12, we see that there is a plateau in performance that is reached at higher epoch counts. In Table 8, we freeze different parts of the model to understand if certain parts of the model are critical for `NEFT`.

Table 7: `AlpacaEval` Win Rate with ChatGPT (GPT-4 in parentheses) evaluator under different decoding strageties from this we can see that there seems to be little variation in performance. The WizardLM and LLaMA-Chat hyperparameters were obtained from generation config files from Huggingface. All sampling techiniques had a reptition penalty of 1.2.

| Hyper. Source | top_p | temp. | LLaMA2-7B (Evolve) | LLaMA2-7B-NEFT (Evolve) |
|---|---|---|---|---|
| Base | greedy | | 62.55 (70.34) | 67.58 (79.60) |
| HP 0 | 0.1 | 0.1 | 63.11 | 66.83 |
| HP 1 | 0.35 | 0.5 | 62.48 | 66.71 |
| WizardLM | 0.6 | 0.9 | 62.05 | 66.96 |
| LLaMA-2 | 0.9 | 0.6 | 63.73 (70.49) | 65.47 |

Table 8: `AlpacaEval` Win Rate according to ChatGPT while varying the set of trainable parameters when finetuning LLaMA-2-7B on the Alpaca dataset. The top two rows have all parameters set as trainable.

| Setting | AlpacaEval (ChatGPT Eval) |
|---|---|
| standard finetuning | 48.26 |
| NEFT | 62.55 |
| NEFT+Embed frozen | 61.06 |
| NEFT+LM-head frozen | 61.12 |
| NEFT+Attention blocks frozen | 22.17 |
| LLaMA-2 (no finetuning) | 22.17 |

Table 9: `NEFT` performs better than FreeLB.

|  | ChatGPT Win Rate |
|---|---|
| LLaMA-1-7B (Evolve) | 62.30 |
| +`NEFT` | 67.45 |
| +FreeLB (after hparam tuning) | 63.48 |

Table 10: Using ChatGPT as the evaluator, we observe a slight performance increase when training with longer sequences on the ShareGPT data compared to standard finetuning at the same sequence length.

| LLaMA-2 (7B) | Split 512 | Split 1024 |
|---|---|---|
| ShareGPT | 63.48 | 61.68 |
| +`NEFT` | 64.22 | 64.35 |

Table 11: Ablating the alpha for LLaMA-1 trained on alpaca for values with powers of 10, we see that for lower values of alpha performance on AlpacaEval does not change, and for very high values like 100, it makes instruction finetuning very difficult.

| Alpha | AlpacaEval (ChatGPT Eval) |
|---|---|
| 100 | 0.00 |
| 50 | 57.52 |
| 10 | 61.99 |
| 5 | 58.32 |
| 1 | 49.69 |
| 0.1 | 47.70 |
| 0 | 48.50 |

### A.3 ADDITIONAL ANALYSIS

**How does the noise impact the tokens?** Since our modeling involves adding random noise to embeddings during the training stage, we examined whether the added noise changes the semantics of the token sequences in the training data. For this analysis, we sample a random 5200 samples from the Alpaca dataset, embed each training point using the embedding layer of different models, and then add different levels of noise by varying the scaling factor $\alpha$. We then project the noised embeddings back to their closest neighbor in the embedding matrix. We compute the % of token flips in each sentence and average the flip rate across all the samples. We present the flip scores for 7 models in Fig. 7 (Left). While none of the sentences had any flips up to $\alpha = 15$, we see some flips when $\alpha \geq 25$. Note that all the results presented in the paper use $\alpha \leq 15$. Interestingly, a LLaMA-1 model finetuned on Alpaca does not show any flips even at higher levels of $\alpha$.

**Impact of noise on embedding similarity** We also analyzed how the similarity of embeddings changes when we perform `NEFT` training. We looked at the top 100 singular values of the embedding similarity matrix for all the models. We present the results in Fig.7 (Right). For a given base model (LLaMA-1 or LLaMA-2), the singular value distribution did not change across the variants, with or without `NEFTune`. This shows that the addition of noise during training does not impact the embedding similarity distribution very much.

**How does noise impact embeddings?** In the previous analysis, we evaluated whether any tokens flipped when noising sequence of real training data. We also examine the embeddings of 3 models, LLaMA-2, LLaMA-2 Alpaca, and LLaMA-2 Evol-Instruct in isolation. In this experiment, we sweep over all of the tokens in the vocabulary adding noise to each one, and count the number of noised embeddings whose nearest neighbor is different from their un-noised starting point. We present the results in Fig. 9. We vary 2 factors: base noise scale $\alpha$, and the "sentence length" scaling factor $L$ which is used in calculating the final noise coefficient. Even at high levels of noise, only a

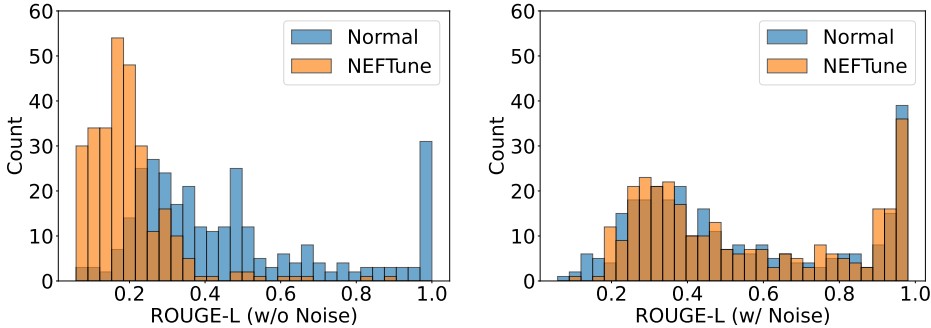

Figure 6: Left shows the ROUGE-L of training with and without NEFTune. Right shows ROUGE-L of training with and without NEFTune with noise added during inference. We can see that a lot of the training data can be recovered during generation when applying noise during inference time. Note the set up similar to Figure 5.

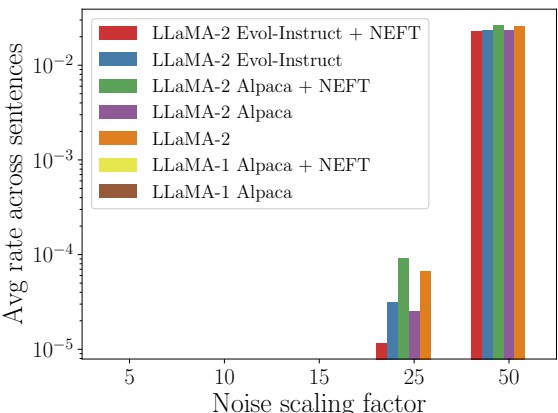

Figure 7: Ratio of tokens per sentence flipped at different levels of noise added to embeddings.

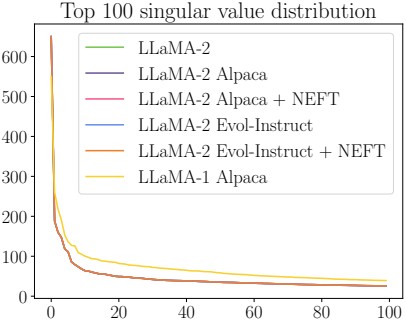

Figure 8: Top-100 eigenvalues of embedding similarity matrices across models.

Table 12: `AlpacaEval` ChatGPT Win Rate as a function of the number of finetuning epochs.

| Epochs | LLaMA-2 (7B) Alpaca | +NEFT |
|--------|---------------------|-------|
| 1 | 40.50 | 55.09 |
| 3 | 48.26 | 62.55 |
| 5 | 48.94 | 62.24 |
| 7 | 48.63 | 60.50 |
| 9 | 47.45 | 58.14 |

Table 13: `AlpacaEval` win rates judged by ChatGPT of different regularizers. We ablated dropout over the attention probabilities over $0.05, 0.1,$ and $0.5$, and weight decay, $0.1, 0.2,$ and $0.5$, reporting the best for each of the techniques. As seen below, the benefits of NEFTune are not seen by the other regularizers.

| Method | NEFTune | Dropout | Weight Decay | Baseline |
|--------|---------|---------|--------------|----------|
| Win Rate | 61.9 | 47.8 | 48.0 | 48.5 |

Figure 9: Number of tokens flipped at different levels of noise and sentence lengths (out of 32000). Model names corresponding to each plot are in the title.

very small number of tokens actually flip ($\leq 0.4\%$). This shows that NEFT training does not change the semantic relationships between the tokens.

Interestingly, this experiment suggests that, if considering NEFTune as a type of data augmentation applied to the embeddings of training tokens, the actual effects of NEFT are mostly a result of perturbations in intra-token space since the noise scales used rarely amount to displacements that transform any individual token into a meaningfully different one. Nevertheless, in Figure 10 we show the tokens corresponding to embeddings of LLaMA-2 Alpaca model that do actually flip even with the lowest amount of noise utilized in our study.

Table 14: $\alpha$ used for Fig 3.

| | Alpaca | Evol-Instruct | OpenPlatypus | ShareGPT |
|-----------|--------|---------------|--------------|----------|
| OPT 6.7B | 15 | 15 | 5 | 15 |
| LLaMA-1 7B | 10 | 10 | 15 | 10 |
| LLaMA-2 7B | 5 | 5 | 15 | 15 |

```
['竹', 'elte', 'canvas', 'мон', '_Introduction', 'adó', '_груп', '_fresh', '<0x6B>', '_pal', 'тів', '_Rio',
'd', '═══', '_chair', '¿', 'ℝ', '_galax', '_klein', 'obre', 'chain', 'hom', '与', 'менталь', '_экс', '_svo',
'_Rab', '}\\)', 'pot', '_DES', 'ctors', '_Santos', '_желез', 'ръ', 'cluded', 'hit', 'ých', '_Дже', '_./', 'д',
'_notion', '完', '_Th', '界', 'adj', 'ichtung', '_stands', '_wieś', '_gepublic', 'ilt', '_∈', '_eggs', 'änd',
'·', 'ɪ', '_veg', 'Insert', '_sure', '』', '_applicable', '_arr', 'Download', '}^\\', '_Branch', 'uses',
'_focus', '_rural', 'ferences', 'categories', '_hidden', 'AV', 'legt', '_Audiod', '_меди', 'hens', '_reject',
'exec', '_entering', '_leta', 'ic', '_estimates', '_bât', 'бин', '_Gel', '_mai', '_hyd', ',%', '})$,',
'_multiple', '_older', '球', '_Tochter', '_Jul', 'accept', 'xpath', 'ándose', 'omed', 'tout', '{$', 'dr',
'_JSON', 'istro', '_deployment', 'êque', '_општини', 'andr', '无', 'idents', 'кт', '_Arag', '_largely']
```

Figure 10: Tokens flipped at $\alpha = 1$ and sentence length = 512 in LLaMA-2

### A.4 QUALITATIVE EXAMPLES

For these qualitative examples for LLaMA-2 finetuned models, we chose one instructions that requires knowledge explanation and one that involves "creatativity." We pull these examples from `AlpacaEval`, particularly the Vicuna Eval subset. Note we did not actively cherry-pick these examples.

---

**AlpacaEval (Vicuna Subset) Instruction**: Can you explain the basics of quantum computing?

**LLaMA-2-Alpaca-7B Response**:

Quantum computing is based on the principles of quantum mechanics, which allow for phenomena such as superposition and entanglement between particles. This means that information can be processed in parallel instead of sequentially like traditional computers. Quantum bits (qubits) are used to store data instead of classical bits, allowing more complex computations to take place with fewer resources than would otherwise be necessary.

**LLaMA-2-Alpaca-7B-NEFT Response**:

Quantum computing is based on the principles of quantum mechanics, which describe how particles such as electrons and photons behave at the subatomic level. Unlike traditional computers, which process information in binary digits (bits) that can be either 0 or 1, quantum bits (qubits) can exist in multiple states between 0 and 1 simultaneously. This phenomenon is known as superposition and allows for much more complex calculations to be performed than would otherwise be possible with classical computers. Quantum algorithms are designed to take advantage of this property by encoding data into qubit states so that it can be processed exponentially faster than conventional methods. Additionally, quantum entanglement–the ability for two particles to remain connected even over great distances–allows for large amounts of data to be sent across networks quickly and securely. The potential applications of quantum computing range from cryptography and artificial intelligence to drug discovery and material science. With continued research and development, quantum technology has the potential to revolutionize many industries.

**LLaMA-2-ShareGPT-7B Response**:

Quantum computers are a type of computer that use quantum-mechanical phenomena, such as superposition and entanglement, to perform operations on data. They have the potential to solve certain problems much faster than classical computers, which are limited by the laws of classical physics.

One way to think about how a quantum computer works is in terms of its basic building blocks or "qubits." Qubits can exist in multiple states at once (a property known as superposition), and they can also be linked together so that their state depends on the state of other qubits (a phenomenon called entanglement). These properties allow quantum computers to perform many calculations simultaneously, taking advantage of the principles of quantum mechanics to achieve exponential speedup for certain types of computations.

**LLaMA-2-ShareGPT-7B-NEFT Response**:

Quantum computers are a type of computer that use quantum-mechanical phenomena, such as superposition and entanglement, to perform operations on data. They have the potential to solve certain problems much faster than classical computers, which are limited by the laws of classical physics.

Here is a brief overview of some key concepts in quantum computing:

1. Superposition: In classical computing, bits can only be either 0 or 1 at any given time. However, in quantum computing, qubits (quantum bits) can exist in multiple states simultaneously, known as superpositions. This allows for many calculations to be performed simultaneously, increasing the speed of computation.

2. Entanglement: When two particles become "entangled," their properties become correlated even when they are separated by large distances. For example, if two photons are created with opposite polarizations, measuring the state of one photon will instantly determine the state of the other, regardless of how far apart they are. This property can be used to create a network of entangled qubits, allowing them to work together to solve

---

complex problems.

3. Interference: Just like waves can interfere with each other to produce new patterns, qubits can also interfere with each other through a process called quantum interference. By manipulating these interferences, quantum algorithms can be designed to take advantage of the unique properties of quantum systems.

4. Measurement: One of the biggest challenges in working with quantum systems is that measurements collapse the wave function, effectively freezing the system into its final state. This means that once a measurement is taken, the information stored in the quantum system cannot be changed anymore. As a result, quantum computations must be designed carefully to ensure that the desired outcome is achieved before the measurement is taken.

5. Error correction: Because of the inherent instability of quantum systems, errors can occur during computation. To overcome this challenge, error correction codes must be implemented to detect and correct mistakes made by the quantum system. These error correction techniques are critical to making practical quantum computers a reality.

6. Topological Quantum Computing: Another approach to building a quantum computer uses topological phases of matter, where the order parameter takes values in a non-commutative space. It has been shown that certain classes of topologically ordered states can host fault-tolerant quantum computation.

7. Degrees of freedom: In classical computing, there is usually just one degree of freedom per bit - whether it is set to 0 or 1. But in quantum computing, there can be multiple degrees of freedom per qubit, leading to more possibilities for encoding information and performing calculations.

8. Classical vs Quantum gates: Gates are basic building blocks of quantum circuits, similar to logic gates in classical computing. While both classical and quantum gates perform operations on data, the rules governing them differ significantly. For instance, while classical gates operate on binary digits (bits), quantum gates operate on qubits, and the principles underlying their behavior are fundamentally different.

**LLaMA-2-Evolve70k**:

Sure, I'd be happy to help!

Quantum computing is a type of computation that uses quantum-mechanical phenomena, such as superposition and entanglement, to perform operations on data. Unlike classical computers, which use bits (which can only have one of two values, either 0 or 1) to represent information, quantum computers use qubits (which can exist in multiple states simultaneously). This allows for much faster processing times when it comes to certain types of calculations.

One way that quantum computers are different from classical computers is through their ability to perform parallel computations. In a classical computer, each bit must be processed sequentially, but in a quantum computer, many qubits can be manipulated at once using something called a quantum gate. These gates allow for complex algorithms to be executed quickly by taking advantage of the properties of quantum mechanics.

Another key aspect of quantum computing is entanglement, which occurs when two or more particles become connected in such a way that the state of one particle affects the state of another, regardless of how far apart they may be. Entangled qubits can be used to perform certain types of calculations much faster than non-entangled ones because the results of measurements made on them are predetermined based on their initial state.

Overall, while there is still much research being done into the potential applications of quantum computing, this new field has great promise for solving some of society's most challenging problems related to cryptography, optimization, and simulation.

**LLaMA-2-Evolve70k-NEFT**:

Sure, I'd be happy to help!

Quantum computing is a type of computation that uses quantum-mechanical phenomena, such as superposition and entanglement, to perform operations on data. Unlike classical computers, which use bits (which can only have one of two values, either 0 or 1) to represent information, quantum computers use qubits, which can exist in multiple states simultaneously. This allows for much faster processing times when it comes to certain types of calculations.

The basic idea behind quantum computing is to use a series of quantum gates to manipulate

the state of individual qubits. These gates are similar to the logic gates used in classical computer architecture, but they operate on a different level of abstraction. For example, instead of using AND and OR gates to build more complex circuits, quantum gates like the CNOT gate allow us to create entangled pairs of qubits with specific correlations between their states.

One of the key advantages of quantum computing is its ability to solve certain problems exponentially faster than classical computers. One famous example is factorization, where we need to find the prime factors of a large number. Using Shor's algorithm, a quantum computer can factorize numbers much larger than what would be possible with classical methods. Another area where quantum computing has shown promise is in simulating quantum systems, which could lead to new insights into materials science and chemistry.

However, there are also many challenges associated with building practical quantum computers. One major issue is error correction, since any disturbance to a qubit can cause it to lose its quantum properties and become a classical bit. Researchers are currently working on developing error-correcting codes that can mitigate these errors and make quantum computations more reliable over time. Additionally, there are still many open questions about how best to architect quantum computers, including issues related to scalability and interconnectivity between components.

**LLaMA-2-OpenPlatypus**:

Quantum computing is a rapidly developing field that has the potential to revolutionize how we process and store information. It relies on the principles of quantum mechanics, which describe the behavior of matter and energy at very small scales, such as atoms and subatomic particles.

In classical computers, bits represent either 0 or 1, but in a quantum computer, qubits can exist in multiple states simultaneously. This property, called superposition, allows quantum computers to perform many calculations in parallel, greatly increasing their processing power. Additionally, quantum computers use entanglement, where two or more particles become connected in such a way that the state of one particle instantly affects the other, regardless of the distance between them. This phenomenon enables quantum algorithms to solve certain problems much faster than classical algorithms.

One example of a problem that could be solved quickly by a quantum computer is factoring large numbers into prime factors. While this may not seem like a significant breakthrough, it would have wide-ranging implications for fields such as cryptography, since it would enable the rapid cracking of codes used to secure sensitive data. Another area where quantum computing could have a major impact is drug discovery, as it could help identify new treatments for diseases much faster than traditional methods.

However, building a practical quantum computer with enough qubits to be useful remains a challenge. Currently, most quantum computers are relatively small and prone to errors due to environmental interference. Researchers are working on improving both the number of qubits and the accuracy of quantum computations, making progress towards realizing the full potential of this exciting technology.

**LLaMA-2-OpenPlatypus-NEFT**:

Quantum computing is a field of computer science that deals with the use of quantum mechanics for computation. It involves developing and studying computational models based on the principles of quantum theory, which describe the behavior of matter and energy at very small scales. The goal of quantum computing is to take advantage of the unique properties of quantum systems, such as superposition and entanglement, to solve complex problems more efficiently than classical computers can.

In classical computing, bits represent either 0 or 1, but in quantum computing, qubits (quantum bits) can be in multiple states simultaneously until they are measured. This property, called superposition, allows quantum computers to perform many calculations in parallel, greatly increasing their processing power compared to classical computers. Another key feature of quantum computing is entanglement, where two or more particles become correlated such that the state of one particle instantly affects the other, regardless of the

distance between them. Entanglement can be used to transmit and process information instantaneously, enabling faster communication and computations.

One of the main challenges in building practical quantum computers is controlling and manipulating individual quantum bits accurately enough to realize useful algorithms. Currently, most quantum computers rely on experimental platforms using trapped ions, superconducting circuits, or photons. These devices are still relatively fragile and prone to errors, known as "noise," which must be corrected by sophisticated error-correction codes. However, significant progress has been made in recent years towards scaling up quantum computers and achieving higher levels of noise tolerance.

Some potential applications of quantum computing include cryptography, optimization problems, simulation of physical systems, and drug discovery. Quantum computers could also play a role in solving some of society's biggest challenges, such as climate change modeling, materials science, and protein folding research. While much work remains to be done before quantum computers become widely available, the field continues to attract attention from both academia and industry due to its great promise for transforming various fields of study and engineering.

---

**AlpacaEval (Vicuna Subset) Instruction**: Write a symphony concert review, discussing the orchestra's performance and overall audience experience.

**LLaMA-2-Alpaca-7B Response**:
The symphony concert was truly remarkable! The orchestra played with great energy and skill, creating a beautiful soundscape for the audience to enjoy. Every piece they performed was expertly executed, from the gentle melodies of Mozart to the powerful climaxes of Beethoven.

The acoustics in the hall were perfect; every note could be heard clearly and distinctly. The lighting design added another layer of emotion to each piece, helping to draw out the emotions within the music. Overall, it was a wonderful evening full of fantastic music and an unforgettable experience for all who attended.

**LLaMA-2-Alpaca-7B-NEFT Response**:
The symphony concert was truly remarkable! The orchestra played with such passion and energy that it brought tears to my eyes. Every note they struck seemed to reverberate through the hall in perfect harmony. It was clear from their performance that each musician had put in countless hours of practice and dedication to create this beautiful sound.

The conductor did an excellent job leading the musicians throughout the evening. His movements were full of emotion as he guided them through the pieces. He would often pause for moments at a time allowing the audience to take in every moment of the piece before moving on. This allowed us all to appreciate the intricacies of each song even more.

The crowd was filled with people of all ages; some who knew the songs well and others who were there to simply enjoy the night. There wasn't one person in attendance who didn't seem to be having a great time. Even during the quieter parts of the show, you could feel the excitement in the air. Whenever something particularly exciting happened everyone cheered and applauded.

Overall, I can say without a doubt that this symphony concert was one of the best experiences I have ever had. From start to finish, everything about it was absolutely amazing. If given the chance, I will definitely go back again next year.

**LLaMA-2-Share-7B Response**:
On Saturday evening at Carnegie Hall, I had the privilege of attending a symphony concert featuring the New York Philharmonic Orchestra under the direction of Maestro Lorin Maazel.

The program for the evening consisted of three works by some of classical music's most renowned composers - Beethoven's "Egmont" Overture, Tchaikovsky's Piano Concerto No. 1 with soloist Lang Lang, and Dvořák's Symphony No. 9 in E minor, Opus 95 ("From the New World").

Maestro Maazel led the orchestra through each piece with precision and passion, drawing out rich and vibrant sounds from every section of the ensemble. The strings played with great depth and emotion, while the woodwinds and brass added colorful accents throughout the evening.

Lang Lang was a particularly impressive performer as well, displaying technical mastery and artistic sensitivity in his interpretation of Tchaikovsky's concerto. His playing was full of energy and expression, and he seemed completely immersed in the music.

As for the audience, it was clear that everyone present was there to appreciate the beauty and power of live orchestral music. There were frequent bursts of applause and cheers throughout the night, and many people around me were wiping tears of joy from their eyes during the final movement of Dvořák's symphony. It was a truly moving experience to be surrounded by so many people who shared a love for this timeless art form.

Overall, the concert was a true triumph for both the musicians on stage and the audience in attendance. It was a reminder of why we continue to support and celebrate live music, and I left the hall feeling inspired and uplifted.

**LLaMA-2-Share-7B-NEFT Response**:
On Saturday night at Carnegie Hall, the New York Philharmonic put on a breathtaking performance of Symphony No. 5 by Ludwig van Beethoven. Under the direction of Music Director Jaap van Zweden, the orchestra delivered a powerful and emotional rendition of this classic work that left the packed house in rapturous applause.

The evening began with a captivating performance of Mozart's Overture to Die Zauberflöte, which set the stage for the grandeur of what was to come. As the lights dimmed and the orchestra launched into the opening notes of the Fifth Symphony, it was clear that this would be a special evening. Van Zweden expertly guided the orchestra through each movement, drawing out every nuance of the music while maintaining a sense of momentum and energy.

One of the standout features of the performance was the orchestra's dynamic range. From the softest pianissimo passages to the loudest fortissimo climaxes, every member of the ensemble played with precision and depth. The brass section, in particular, shone throughout the evening, delivering rich and full-bodied sound that filled the hall.

In addition to the superlative playing of the musicians, the acoustics of Carnegie Hall were also a major factor in the success of the evening. This iconic venue has a warm and intimate atmosphere that allows listeners to fully immerse themselves in the music. Every note seemed to hang in the air, enveloping the audience in a sonic embrace that made it easy to forget the outside world and lose oneself entirely in the beauty of the music.

As the final chords of the symphony faded away, the audience erupted into thunderous applause, showering the musicians with praise and admiration. It was a truly magical moment, one that will stay with me long after the evening came to a close. Overall, I would say that this was a once-in-a-lifetime experience that any lover of classical music should not miss.

**LLaMA-2-Evolve70k Response**:
I recently attended a symphony concert at the renowned Carnegie Hall in New York City. The evening's program featured works by Beethoven, Mozart, and Tchaikovsky, performed by the prestigious New York Philharmonic Orchestra under the baton of maestro Alan Gilbert.

The concert began with Beethoven's lively "Leonore" Overture No. 3, which set the tone for the evening's festivities. The orchestra played with great energy and precision, showcasing their mastery of the classical repertoire. Maestro Gilbert conducted with grace and fluidity, drawing out each note and phrase to maximum effect.

Next up was Mozart's Symphony No. 29, a beautiful work that demonstrated the composer's deftness with melody and harmony. The orchestra played with even greater nuance than before, creating a rich tapestry of sound that transported me to another world. I particularly enjoyed the woodwind section, whose delicate phrases added depth and dimension to the piece.

Finally, we were treated to Tchaikovsky's iconic Symphony No. 5. This was undoubtedly

the highlight of the evening, as the orchestra brought all of their skill and passion to bear on this towering masterwork. From the haunting opening theme to the triumphant finale, every moment of the performance was filled with emotion and intensity. It was truly a thrilling experience to witness such accomplished musicians bring these timeless compositions to life.

Overall, the evening was a resounding success. The New York Philharmonic Orchestra proved themselves to be among the finest ensembles in the world, playing with technical prowess and artistic sensitivity. Maestro Gilbert showed himself to be a skilled conductor, guiding the orchestra through each work with care and attention to detail. And the acoustics of Carnegie Hall itself were simply breathtaking, providing a perfect backdrop for the music. If you have the opportunity to attend a symphony concert, I highly recommend it - you won't be disappointed!

**LLaMA-2-Evolve70k-NEFT Response**:

As I took my seat in the concert hall, anticipation was building as the lights dimmed and the orchestra began to play. The first piece on the program was Beethoven's Symphony No. 1, which immediately captured my attention with its powerful melodies and intricate rhythms. The conductor led the orchestra with precision and enthusiasm, and the musicians responded with a dynamic performance that showcased their exceptional skill and passion for music-making.

The second movement, marked "Andante Favori," was particularly beautiful, with delicately played strings and woodwinds creating a serene and introspective mood. The third movement, "Allegro," was full of energy and drive, with the brass section adding a bold and assertive presence to the music. The final movement, "Alla Da Prestissimo," was a thrilling conclusion to the symphony, with lightning fast tempos and dazzling virtuosity from all members of the orchestra.

Throughout the performance, I was impressed by the cohesion and unity of the ensemble, as well as the individual excellence of each musician. The sound they produced was rich and full, yet also crystal clear and precise, and there were many moments where I was left breathless by the sheer beauty and power of the music.

In addition to the excellent playing, another highlight of the evening was the warm and welcoming atmosphere created by the orchestra and conductor. They interacted with the audience throughout the performance, sharing insights into the music and encouraging us to enjoy the journey along with them. This made for a truly engaging and immersive experience, one that left me feeling inspired and uplifted long after the last note had faded away.

Overall, this symphony concert was a remarkable event that I will remember for years to come. The orchestra's superb performance combined with the audience's enthusiastic support to create a magical night of music-making that I would recommend to anyone looking for a truly unforgettable musical experience.

**LLaMA-2-OpenPlatypus**:

Title: A Night of Enchantment with the Boston Symphony Orchestra
Date: March 15th, 2023
Venue: Symphony Hall, Boston MA
Review by John Doe for The Boston Globe

The Boston Symphony Orchestra (BSO) returned to its home stage at Symphony Hall last night after a two-week hiatus, and they did not disappoint as they presented their first program under the baton of Music Director Andris Nelsons since his recent appointment extension was announced. With a captivating lineup featuring works from three different eras and composers, this evening proved to be nothing short of enchanting for both musicians and audiences alike.

Opening the show was Jean Sibelius' iconic "Finlandia," which set the tone for what would prove to be an exquisite journey through various musical styles throughout history.

Performed without intermission, the BSO immediately captured our attention with their impeccable precision and dynamic range - particularly during the stirring brass passages that brought us into the heart of Finnish nationalism. It was clear from the outset that Maestro Nelsons had prepared his ensemble meticulously; every note seemed carefully considered yet effortlessly executed, creating a seamless blend between soloists and full orchestra.

Next up was Pyotr Ilyich Tchaikovsky's beloved Violin Concerto in D Major, performed by one of today's most acclaimed violinists, Leonidas Kavakos. From the very first notes, it became apparent why he has earned such widespread praise; his technical prowess was on full display while also exhibiting immense depth of feeling throughout each movement. His partnership with the BSO was truly special, resulting in moments of pure magic when all eyes were fixed upon him as he soared above the orchestral accompaniment.

After intermission came another treat: Gustav Mahler's Symphony No. 4 in G Major ("Tragic"). This work often receives less attention compared to some other Mahler symphonies due to its relative brevity (approximately half an hour), but make no mistake - there is much beauty and emotion packed within these few minutes. Under Mr. Nelsons' guidance, we experienced every twist and turn along the way, culminating in a thrilling climax that left us breathless before suddenly fading away into silence.

As if that weren't enough, the encore offered something entirely unexpected: Samuel Barber's Adagio for Strings. Originally composed as part of a string quartet arrangement, this haunting melody has since become synonymous with mourning and loss thanks largely to its inclusion in films like Platoon and The Elephant Man. Hearing it live provided goosebumps galore as it filled Symphony Hall from top to bottom; even those who may have been unfamiliar with the piece prior could not help but feel moved by its poignancy.

In conclusion, tonight's performance demonstrated exactly why the Boston Symphony Orchestra remains one of the world's preeminent ensembles nearly two centuries after its founding. Each musician appeared fully invested in their craft, delivering stunning interpretations of timeless classics alongside more contemporary pieces. As someone lucky enough to witness this magical event firsthand, I can confidently say that it will remain etched in my memory forever - just like many others who attended last night's sold-out show.
**LLaMA-2-OpenPlatypus-NEFT Response**:
Title: A Night of Enchantment with the Boston Symphony Orchestra

The Boston Symphony Orchestra (BSO) presented their latest concert at historic Symphony Hall in Boston on February 1st, 20XX. The evening was filled with captivating music from some of the most renowned composers throughout history. With conductor Andris Nelsons leading the way, this night proved to be one of enchantment for both performers and audiences alike.

As we took our seats inside Symphony Hall, anticipation filled the air as people chatted excitedly about what they were expecting from tonight's show. The hall itself is quite impressive; its ornate design featuring intricate details like marble columns and stained glass windows creating an atmosphere of grandeur perfect for such an event. Once everyone had found their seat, there was complete silence except for the occasional cough or shuffling of feet until finally, the lights dimmed signaling that it was time for the show to begin.

The first piece performed was Jean Sibelius' "Finlandia," Opus 26, which set the tone for the rest of the evening – grandiose yet serene. As expected, Andris Nelsons brought out each note vibrantly while still allowing space for individual instruments to shine through. It was clear from his direction that he had prepared extensively for this performance, ensuring every aspect of the music came alive under his baton.

Next up was Pyotr Ilyich Tchaikovsky's iconic Violin Concerto in D Major, Opus 35, featuring soloist Gil Shaham. From the very first notes played by Shaham, it became

apparent why he has earned critical acclaim worldwide – his technical prowess combined with emotional depth made for an unforgettable performance. He seamlessly transitioned between different styles and tempos, keeping us enthralled throughout the entirety of the concerto. Whenever he stepped back into the ensemble after his solos, you could hear a collective sigh of relief as if we had all been holding our breath waiting for him to return. Andris Nelsons conducted flawlessly alongside him, adding even more depth and beauty to this already beautiful work.

After intermission, the BSO returned to the stage for Johannes Brahms' Symphony No. 4 in E Minor, Opus 98. This piece marked another highlight of the evening, demonstrating once again how well these musicians can come together as one cohesive unit. Each section had its moment to shine without ever overpowering the others, resulting in a harmonious blend of sound that left us spellbound. Andris Nelsons showed great versatility here too, expertly navigating through various sections while maintaining momentum throughout the whole movement.

To end the night on a high note, the BSO closed with Gustav Mahler's Symphony No. 1 in D Major ("Titan"). Right from the opening bars, it was evident that this would be no ordinary performance. There was an energy within the orchestra that seemed to infect those watching, causing many people around me to clap along enthusiastically despite not having reached the final movement just yet! Andris Nelsons continued to lead them effortlessly, guiding us through this epic journey full of emotion and passion. By the time we reached the climactic finale, cheers erupted from the crowd as if we had all been released from some sort of magical trance. It truly felt like we had witnessed something special, something that will stay with us long after the last note had died away.

In conclusion, the Boston Symphony Orchestra put on an incredible show last night, displaying impeccable musicianship and artistry throughout their performances. Their dedication to bringing new life to classic works while also exploring lesser-known gems was greatly appreciated by all who attended. If you have the opportunity to catch one of their future shows, do not hesitate - allow yourself to become enchanted by their magic!

Here are a couple more examples from the `NEFTune` models that performed the best on AlpacaEval (LLaMA-2 trained on Evolve-Instruct).

**AlpacaEval (Vicuna Eval) Instruction**: What if Vincent van Gogh had been a successful artist during his lifetime?
**LLaMA-2-Evolve70k**:
As an AI language model, I don't have personal beliefs or opinions but based on historical facts and researches, it is believed that Van Gogh would not have experienced mental illness as he was recognized for his artistic talent in his early years before developing psychiatric problems later in life. He may have continued to create more masterpieces and become one of the most famous artists of all time. His legacy might also be different from what we know today since he did not achieve recognition until after his death when his brother sold some of his paintings at auction.
**LLaMA-2-Evolve70k-NEFT**:
As an AI language model, I don't have personal beliefs or opinions but here are some possible scenarios based on historical facts and speculation:
1) Van Gogh would likely have continued to create art throughout his life, as he was prolific in his output even before his breakthrough success. However, with financial stability and recognition, he may have experimented more with different styles and themes, pushing himself creatively in new directions.
2) His relationships with others might also have changed. With money and fame comes social status, which could have opened up opportunities for him to mingle with other artists, writers, and intellectuals of his time. He may have formed deeper friendships and collaborations with fellow creatives, potentially leading to further innovation within the arts community.

3) Unfortunately, mental illness can be difficult to treat, especially in the late 19th century when medical knowledge was limited. Despite his success, it's unlikely that Van Gogh's struggles with depression and anxiety would have disappeared. In fact, they may have intensified due to the added pressure and expectations placed upon him by society and his peers.

4) Ultimately, whether or not Van Gogh became a successful artist during his lifetime wouldn't necessarily change the impact he made on modern art history. His work has endured through countless exhibitions, books, films, and reproductions, cementing his legacy as one of the most influential painters of all time.

**AlpacaEval (1st) Instruction:** What are the names of some famous actors that started their careers on Broadway?

**LLaMA-2-Evolve70k:**

1. Meryl Streep
2. Robert De Niro
3. Al Pacino
4. Kevin Spacey
5. Nathan Lane
6. Audra McDonald
7. James Cagney
8. Bette Midler
9. Richard Gere
10. John Lithgow

**LLaMA-2-Evolve70k-NEFT:**

1. Al Pacino - He made his stage debut in 1967's "Does a Tiger Wear a Necktie?"" and went on to star in several other Broadway productions, including "The Indian Wants the Bronx"" (1968), "The Resistible Rise of Arturo Ui" (1969) and "Serpico" (1973).

2. Meryl Streep - She began her career on Broadway in 1975 with the play "Julia," followed by roles in "Lost in Yonkers" (1991), "A Little Night Music" (2009) and "Three Tall Women" (2018).

3. James Earl Jones - The actor first appeared on Broadway in 1957's "The Long Dream," going on to star in numerous plays and musicals throughout his career, including "Of Mice and Men" (1974), "Fences" (1987) and "Driving Miss Daisy" (1987).

4. Nathan Lane - He made his Broadway debut in 1985's "A Funny Thing Happened on the Way to the Forum," winning a Tony Award for his performance. Other notable shows include "The Producers"" (2001), "The Nance" (2013) and "Angels in America" (2018).

5. Bette Midler - The singer-actress got her start on Broadway in 1965's "Fiddler on the Roof," later earning critical acclaim for her performances in "Forbidden Planet" (1989), "Golden Age" (2005) and "I'll Eat You Last: A Chat With Sue Mengers" (2013).

Here is a case where both models from LLaMA-2 trained on Evolve-Instruct cannot output the correct answer:

**AlpacaEval (Vicuna Eval) Instruction:** Given that f(x) = $5x^3$ - 2x + 3, find the value of f(2).

**LLaMA-2-Evolve70k:**

f(2) = $5(2)^3$ - 2(2) + 3
= 5 * 8 - 4 + 3
= 19 - 6 + 3
= 22

**LLaMA-2-Evolve70k-NEFT:**

To find the value of f(2), we need to substitute x=2 into the function and simplify it. We have:
f(2) = $5(2)^3$ - 2(2) + 3
= 40 - 8 + 3
= 45
Therefore, f(2) = 45.

