# OpenReview forum: "NEFTune: Noisy Embeddings Improve Instruction Finetuning"
_ICLR.cc/2024/Conference — ICLR 2024 poster_

### Official Review · Reviewer_kT9q · 2023-10-27

**Soundness:** 3 good
**Presentation:** 3 good
**Contribution:** 3 good
**Rating:** 6
**Confidence:** 4

**Summary:**

This paper introduces a method for fine-tuning large language models. The authors propose an extremely simple modification to the standard procedure, adding noise to the embedding vectors during fine-tuning. Their method is called NEFTune. The authors experiment with several language models and benchmarks, showing strong results across the board.

**Strengths:**

This paper has many strengths.

1. For starters, the simplicity of the method is hard to understate. Researchers and practitioners can try this with only a few lines of code. This is very valuable and a great strength of this paper.
2. The experiments show strong results across the board. The authors study many models and datasets, and ubiquitously see large gains.
3. Fine-tuning large language models is a vibrant research direction, and as a paper that advances our understanding and capabilities in such enterprise, I believe this would be of interest to many in the community.
4. There are many ablations in this work that are informative to the readers.
5. The paper is clear and well written

**Weaknesses:**

1. All experiments in this paper are done with autoregressive models. Studying other kinds of models (e.g. BERT, T5) would be a valuable addition to this paper, since these models are still used for many downstream applications today.
2. The authors don't present error bars in the experiments, which . Fine-tuning language models can be notoriously noisy (e.g. [1]), and precisly understanding the magnitude of the noise in the presented experiments would be very valuable.
3. There is little analysis on scaling trends. Despite some experiments with QLORA, almost all experiments in the paper are conducted with 7B parameter models. Showing that the gains from the proposed method do not diminish vanish with scale would greatly strengthen the paper, as it would demonstrate it's potential for larger models.
4. There is little discussion on hyper-parameter tuning. It would be great if authors would show that the comparisons are fair but ensuring the computational budget spent for tuning hyper-parameters for the baseline and their method is equal. One concern is that the baseline is more poorly tuned that their method.

[1] Dodge, Jesse, et al. "Fine-tuning pretrained language models: Weight initializations, data orders, and early stopping." arXiv preprint arXiv:2002.06305 (2020).

**Questions:**

1. Do the authors think this method would also work for pre-training?
2. In Table 3, do the numbers next to the method name correspond to the value of alpha? If so, I'm surprised to see such a big variance, and also surprised by the non-convex behavior in many cases. Perhaps this also ties with weakness #2.
3. This is not a question, but I just wanted to thank the authors for starting their bar plots at zero. It is refreshing not to be visually misled.

---

> ### Author Response · Authors · 2023-11-16
> **Response to Reviewer kT9q**
>
> Thank you for your time, Reviewer kT9q.
>
> > Table 3, do the numbers next to the method name correspond to the value of alpha?
>
> Yes, in Table 3 the number next to the method name is the corresponding alpha. Unlike the QLORA setting, we do not see the big variance for the Alpaca dataset under full parameter tunning from Table 6 different alphas have a similar increase in performance. Additionally, for AlpacaEval, we see a Std Err of less than 2% for all models studied.
>
> > There is little analysis on scaling trends.
>
> We added a $70$B model to Table 2, finding NEFTune increases performance at this scale as well. Using the hyperparameters for supervised finetuning from the LLaMA-2 paper, we finetuned LLaMA-2 70B on Evolve-Instruct, finding that with NEFTune performance increased by about 10%. This illustrates that at larger scales this method can be applied.
>
> > Hyperparameter Tuning
>
> We did not engage in extensive hyperparameter tuning for every model and dataset instead relying on fixed hyperparameters. However, we did do an initial hyperparameter tuning for the Alpaca model where we improved on the reported model as per the leaderboard from ~26% to ~32% (Adding NEFT puts the performance at ~62%).
>
> > BERT, T5 models
>
> Work like FreeLB, which was compared and inspired NEFTune, has been known to improve model performance on BERT classification tasks--like GLUE. We suggest users use this work and its subsequent works for improving classification tasks, which is well-studied.
>
> > Do the authors think this method would also work for pre-training?
>
> This is a good question. This is difficult to tell as the tasks are different, and we prefer not to speculate on pretraining.
>
> Please let us know if you have any more questions.

---

> > ### Comment · Reviewer_kT9q · 2023-11-21
> >
> > I would like to thank the authors for responding to some of my comments. Since some of my concerns remain, I'm sticking to my original score.

---

> > > ### Author Response · Authors · 2023-11-22
> > > **Response to Reviewer kT9q**
> > >
> > > Thank you for your engagement, kT9q.
> > >
> > > Concerning your question about error bars, our runs for LLaMA-1 training on Alpaca with $\alpha=5$ with five different seeds have now finished. We scored these runs on AlpacaEval with the ChatGPT Evaluator, and, at the moment, the win rates for the individual trials are $60.50$, $60.00$, $58.32$, $59.07$, and $59.35$ for a mean result of $59.86$. The consistency between runs (standard error is $0.34$) suggests that similar results reported in the main work are likely stable with respect to random seeds.

---

### Official Review · Reviewer_nFQe · 2023-10-29

**Soundness:** 2 fair
**Presentation:** 3 good
**Contribution:** 2 fair
**Rating:** 5
**Confidence:** 4

**Summary:**

The authors demonstrate that adding noise to the embedding vectors during finetuning leads to significant performance increase in large language models. The results were demonstrated using two evaluation benchmarks, AlpacaEval and OpenLLM Leaderboard tasks. Only the former demonstrated significant increase where the NEFT models increased the output length while preserving diversity.

**Strengths:**

- The authors demonstrated empirically that a simple regularization of adding uniform noise to the embedding features lead to an average improvement of 15% across 5 datasets.
- The performance gains were demonstrated using 3 different LLMS, which suggests the generalizability of the method to other architectures.

**Weaknesses:**

- Overgeneralizing performance gains: While the authors demonstrated significant performance gains using the AlpacaEval dataset with 805 instructions, the NEFTune model did not show significant improvements on the OpenLLM Leaderboard tasks (Fig 3), which is more critical since it tests for reasoning and truthfulness. The authors' claim should consider this lack of improvement in their abstract and title so as to not increase the hype of this method without strong foundations.


- Contribution of uniform noise is unclear: The authors clarified that longer outputs caused AlpaceEval scores to increase by 16%. To demonstrate that their method did not cause spurious outputs, they evaluated the model output in terms of 2-gram, log-diversity and whitespace lengths (Table 4). I am curious to know how the NEFT method increases output length without increasing 2-gram repetition and maintaining diversity. Is the model output riddled with sentence fillers, and results for 4-gram analysis were not included in the main text? The authors provided a qualitative example of the NEFT output. I am curious to know if this was a specifically chosen sample. The authors conducted human studies of which human annotators preferred the non NEFT in 30 instances (140-(80+22)). It will be helpful to show the output where the human annotators preferred the non-NEFT output over the NEFT ones. Additionally, I do not think Gaussian Noise 5 should be directly compared against Uniform Noise 5? Perhaps other forms of noise distributions should be considered to understand what the uniform noise is doing.

- Need more clarity on computation:  As the authors mentioned in the Conclusion & Limitations, it is unclear why NEFT works on the AlpacaEval and not on the OpenLLM Leaderboard. More evaluation is needed on why there is a huge disparity in performance gains between metrics. Furthermore, a framework or theory on why the uniform noise improves AlpacaEval metric should be explored. Plotting the training and test loss distribution does not add much to the understanding of the computation. A more thorough analysis of why adding uniform noise seemingly improves performance needs to be explored.

**Questions:**

- I am curious to know why are there no performance gains on the OpenLLM Leaderboard tasks?
- Are there any other forms of regularization at the embedding level that replicates this result?

---

> ### Author Response · Authors · 2023-11-16
> **Response to Reviewer nFQe Part 1/2**
>
> Thank you for your time, Reviewer nFQe!
>
> > Overgeneralization of Results -- No performance gains on the OpenLLM Leaderboard tasks
>
> While we do already discuss a lack of improvement on the leaderboard tasks, we have added an additional line in the abstract itself that states that NEFTune does not seem to improve performance on OpenLLM Leaderboard tasks like MMLU. However, it is important to note two things about the nature of the leaderboard test datasets and the LM-Eval-Harness setup.
>
> Firstly, the rankings suggest that OpenLLM Leaderboard performance is largely dependent on the base model and the relevance of the finetuning dataset. It is unclear how a regularization technique like NEFT (applied to the same base model on the same FT dataset) might impact this process, which is why we approached the question from an empirical standpoint and simply report our findings.
>
> A second important detail that might offer more of a mechanistic explanation, is that the OpenLLM Leaderboard tasks are measured by computing the loglikelihood on each of the candidate answers to the question. Then we check whether the model assigned the highest likelihood to the ground truth answer. This is a standard methodology, originally proposed by the GPT3 paper, and the one used to produce all the results on the leaderboard. However, since this set of evaluations is therefore not _generative_ the model's propensity to generate more thorough and detailed answers to conversational requests, as showcased in AlpacaEval, probably doesn't factor into the model's performance on these tasks. However, we report these results clearly in the main work because we felt it important to show that performance on these evaluations was not negatively impacted under our intervention.
>
> > Are there any other forms of regularization at the embedding level that replicates this result?
>
> We found that other techniques that manipuliate the embedding space (i.e FreeLB) improve AlpacaEval performance as well. See Table 9 in the Appendix.

---

> > ### Author Response · Authors · 2023-11-16
> > **Response to Reviewer nFQe Part 2/2**
> >
> > > To demonstrate that their method did not cause spurious outputs, they evaluated the model output in terms of 2-gram, log-diversity and whitespace lengths (Table 4). I am curious to know how the NEFT method increases output length without increasing 2-gram repetition and maintaining diversity. Is the model output riddled with sentence fillers, and results for 4-gram analysis were not included in the main text?
> >
> > Table 4 in the main text aims to provide a concise summary of the analysis of the effect of NEFT on generation length and repetition characteristics of the model and data pairings we studied. While only 2-gram repetition is shown, the log diversity measure incorporates 2-gram, 3-gram, and 4-gram repetition rates into its aggregate calculation to summarize the overall level of token uniqueness. An important detail is that to make a fair comparison, we must measure repetition and diversity on fixed length chunks of text, since _all_ english text will incur a slight increase in repetition as a function of increased length due to the presence of stopwords and other repeated fragments required to write grammatical sentences.
> >
> > We invite the reviewer to look at Appendix A.4 where we compiled a large list of examples. We believe that the responses from the NEFTuned models are no more obviously repetitive or "full of filler" than the normally trained model's reponses, rather they are more detailed and complete. This is of course a subjective matter, which is why we ran a small human study to provide a more quantitative form of evidence as to whether there were any latent quality issues with the Neftune responses that the GPT-based AlpacaEval process might have missed.
> >
> > With all that said, for transparency, we retabulate the first two columns of Table 4, alongside their equivalent measurement _without_ the normalization procedure of taking a fixed length chunk from both the base model and NEFTuned model outputs. Intuitively, what we see is that the NEFTune model responses, which are 2-3x longer, do show a slight uptick in repetition and a decrease in overall diversity as compared to the base model. However, critically, there is no significant systematic difference when you fairly consider the same fixed length subsequence from both models. Restated, we found that for a given number of generated tokens, NEFTune does not cause any significant decrease in token diversity, nor does it increase token diversity for that matter.
> >
> >
> > | Metric/Data | Alpaca (50 toks) | WizardLM-70k (100 toks) | Alpaca (full) | WizardLM-70k (full) |
> > | ---------------------- | -------- | -------- | -------- | -------- |
> > | Base **2-Gram Rep.**       | 1.49     |  3.87    | 1.78     | 4.82     |
> > | NEFT **2-Gram Rep.**       | 1.72     |  3.79    | 4.21     | 6.49     |
> > | Base **Log Div.**          | 15.97    |  10.65   | 15.21    | 11.14    |
> > | NEFT **Log Div.**          | 16.41    |  10.77   | 10.14    | 8.34     |
> > | Base **Whitespace Length** | 60.5     | 138.99   | 60.5     | 138.99   |
> > | NEFT **Whitespace Length** | 169.36   | 225.56   | 169.36   | 225.56   |
> >
> >
> > > NEFTune vs non-NEFTune
> >
> > During human evaluation, NEFTune model outputs were preferred when non-NEFTune model was not comprehensive or long enough. NEFTune model outputs were also preferred when outputed formatting was easier to read (like bullet points). Non-NEFTune models were preferred when the NEFTune model would have details that were not necessary. The annotations were attached in the supplementary material in the original submission.
> >
> > > Contribution of uniform noise is unclear -- Additionally, I do not think Gaussian Noise 5 should be directly compared against Uniform Noise 5?
> >
> > We chose the best alpha for gaussian noise for the paper. However, we have added the additional results we had for other values of gaussian noise. For every value of alpha, it seems that uniform performs better.
> >
> > | Noise Type | Alpha | AlpacaEval (ChatGPT Evaluation) |
> > | --------        | -------| -------- |
> > |     Baseline    | --     | 48.26     |
> > |     Guassian    | 5      | 60.93     |
> > |     Guassian    | 10     | 60.37     |
> > |     Guassian    | 15     | 60.00     |
> > |     Uniform     | 5      | **62.55**     |
> > |     Uniform     | 10     | 61.18     |
> > |     Uniform     | 15     | 61.86     |
> >
> > Please let us know if you have any more questions.

---

> > ### Public Comment · ~Arjun_Singh1 · 2023-11-18
> > **Results on OpenLLM Leaderboard**
> >
> > Hi,
> > I was searching the model on OpenLLM Leaderboard but could not find with name NEFTune.
> > Please can you let me know how to see it on OpenLLM Leaderboard (or its not available there or I am searching at wrong place ?)?
> >
> > https://huggingface.co/spaces/HuggingFaceH4/open_llm_leaderboard

---

> ### Author Response · Authors · 2023-11-19
> **Repsonse to Arjun Singh**
>
> Hello, Arjun. The numbers can be found in the paper, and we will upload them after this work is de-anonymized.

---

> > ### Public Comment · ~Arjun_Singh1 · 2023-11-22
> >
> > Thanks !

---

> ### Comment · Reviewer_nFQe · 2023-11-21
>
> I would like to thank the authors for addressing my comments. It is still unclear to me why NEFTune differs from any other regularization method. The current manuscript does not offer a theory, conjecture or analysis to give insights on why this specific regularization method  works. I have revised my score to 5 but not more than that as I feel the manuscript does not add to our understanding of why NEFTune works.

---

> > ### Author Response · Authors · 2023-11-22
> > **Thank you Reviewer nFQe**
> >
> > Thank you, Reviewer nFQe for your engagement during the review process.

---

> ### Public Comment · ~Arjun_Singh1 · 2023-11-22
> **Is it possible to compare with similar methods which inspired this work ?**
>
> As authors mentioned, the works similar to other existing works. Do you think its possible to compare with those works like FreeLB method by Zhu et al. or by Kong et al. Just wanted to understand if you have done that comparison or does it make sense to have such comparison ? Or may be I missed something ?
>
> "It should be noted that noisy inputs have been used to improve models in various ways. The first
> instance of noise being used to improve language models was the FreeLB method by Zhu et al.
> (2019), who observed that adversarial perturbations boosted the performance of MLM models. The
> noise in this case is not random, but is rather computed by first adding a small Gaussian perturbation
> to the embeddings and then using a gradient step to find the perturbation that maximally alters model
> performance. This adversarial augmentation approach also improves model performance on graphs
> Kong et al. (2022). While our proposed scheme is non-adversarial, we adopt the noise scaling rules
> from these works."

---

### Official Review · Reviewer_Agi9 · 2023-10-31

**Soundness:** 3 good
**Presentation:** 3 good
**Contribution:** 3 good
**Rating:** 6
**Confidence:** 3

**Summary:**

This paper proposes NEFTune, an augmentation strategy for instruction fine-tuning language models.  NEFTune augments samples by adding uniform random noise to the embeddings.  Fine-tuning language models with NEFTune helps models generate longer responses leading to better performance on OOD conversational and instruction eval tasks without harming knowledge evaluated through Q/A tasks. The authors further suggest that NEFTune performs well by reducing overfitting on fine-tuning datasets.

**Strengths:**

* The main strength of this paper are its strong results on improving conversational ability.  Authors show that across 5 different eval datasets, NEFTune improves performance by 8-35%, while retaining performance on knowledge and reasoning Q/A tasks. The proposed approach is simple to implement requiring only a simple change to add noise to the embeddings of the LLM.  This makes it so that the authors can easily test the method on large models (7B) and varied datasets.

* The authors have also made efforts to demonstrate the effects of adding noise to the embeddings in particular leading to the model generating more text, and leading to higher loss (reduced overfitting).

* Augmentations in NLP for training LLMs are relatively unexplored.  The proposed method is similar to prior work in vision, but new to LLM training which typically only trains for a small number of epochs to avoid overfitting.

**Weaknesses:**

* While performance looks strong with NEFTune, the authors do not evaluate with other augmentation strategies such as those in https://arxiv.org/abs/2106.07499.  These methods can help with limited datasets and overfitting.  NEFTune can appear stronger with comparison to other augmentation and regularization strategies.

* Experiments are done only with 7B parameter models - primarily LLAMA-2 7B.  While results appear strong, and are applicable to large models, these models are trained on large amounts of data and have learned good embedding spaces.  It will be interesting to know if results with NEFTune scale with model size.  Does NEFTune work with small models and larger models? Particularly for smaller models, where the embedding space may not be as strong, or the model is not as susceptible to overfitting.

* Figure 4 shows that Alpaca loss increases when training with NEFT as this adds noise to the embedding space.  Does this mean that performance will be worse on Alpaca eval?

**Questions:**

Q1: Given that models have access to the same information, but tend to generate more text, there are some concerns around hallucination the model might be doing.  Have the authors checked this?

Q2: Does NEFTune improve smaller models such as GPT-2?

Q3: Following on Figure 4, is it possible to measure Alpaca training loss and Alpaca validation loss difference? If the model is overfitting and NEFTune reduces overfitting, we should see this discrepancy decrease.  It appears not to be an overfitting argument but smoothing the embedding space where data from the OOD dataset like Evol-Instruct may be.

---

> ### Author Response · Authors · 2023-11-16
> **Response to Reviewer Agi9**
>
> Thank you for your time, Reviewer Agi9.
>
> > Other augmentation techinques
>
> We did try BPE-dropout which is a subword regularization (https://aclanthology.org/2020.acl-main.170.pdf) in addition to NEFTune and FreeLB (a robust optimization technique). We found that BPE-dropout did not improve the performance of the model with similar or slightly lower performance, but FreeLB did improve performance. Table 9 shows the FreeLB vs NEFTune differences on Evolv-Instruct. For BPE-dropout, we performed initial experiments on Alpaca dataset. Is there a particular one you would like us to run?
>
> Setting| Alpaca | Alpaca+BPE |  Alpaca+FreeLB | Alpaca+NEFTune|
> |-----| -------- | -------- | --------        |----------|
> |AlpacaEval (ChatGPT Evaluator)| 48.5     | 44.16     | 58.5     |    61.7    |
>
> > Hallucination
>
> As the the model generates more text, the model has more chances to hallucinate. We find that this is true even for NEFTune models. NEFTune models does not help limit the hallucination problem of models.
>
> > More Model Sizes
>
> We added a $70$B model to Table 2, finding NEFTune increases in perform at this scale as well. Using the hyperparameters for supervised finetuning from the LLaMA-2 paper, we finetuned LLaMA-2 70B on Evolve-Instruct, finding that with NEFTune performance increased by about 10%. This illustrates that at larger scales this method can be applied. We did try to train OPT-1.3B; however, the baseline model outputs degenerated often to the point where evaluating using ChatGPT would result in the flagging OpenAI's content filters too often (https://github.com/tatsu-lab/alpaca_eval/issues/162). Additionally, examining the AlpacaEval Leaderboard, there are almost no instruction models at this scale, which might indicate that 7B+ is really where instruction finetuning has been studied.
>
> > Figure 4 -- Loss/Embedding Smoothing
>
> This is a good question. We did freeze the embedding layer (Table 8) and allowed the model to train; however, we found equivalent boosts even when freezing the word embedding layer. This would indicate there NEFTune is doing more than just smoothing the embedding. Additionally, we unfortunately used the entire training dataset as per the original Alpaca model, so we do not have a validation split from Alpaca for the models we trained. Additionally, we see that models like OPT-6.7B, which is much weaker model than LLaMA-1 and LLaMA-2, and thus may have a weaker embedding space, that performance on AlpacaEval still increases. Nevertheless the we will add an additional experiment showing the train/eval loss curves during training in a future version of the paper.
>
> Please let us know if you have any more questions.

---

> > ### Comment · Reviewer_Agi9 · 2023-11-22
> > **Response to author comments**
> >
> > I would like to thank the authors for addressing many of my comments in the review.  The authors have addressed some concerns with comparison to existing regularization approaches, and addressed the issues with training smaller models.  However, there are still outstanding issues with why significant gains are achieved from NEFTune, and there are not thorough enough measures to fully claim there's no cost to the proposed approach.
> >
> > In particular, there is still outstanding concern that the model is primarily only generating more text, which does not increase knowledge capability, and downstream zero-shot performance on MCQ/retrieval, but may increase on longer-form generation tasks.  As the authors have acknowledged this can also lead to negative effects it the model generates more incorrect information/hallucinations.

---

> > > ### Author Response · Authors · 2023-11-22
> > > **Response to Reviewer Agi9**
> > >
> > > Thank you for your response. We're glad that we were able to address many of your comments.
> > >
> > > There are two primary reasons we believe NEFTune does not affect performance on the four tasks in the OpenLLM Leaderboard. One, they are knowledge-intensive multiple choice tasks and there is no obvious mechanism in which NEFTune or another regularizer would change the knowledge contained in the LLM as this is mostly obtained in the pretraining stage rather than the instruction finetuning phase. Two, the established method for scoring models on this battery of multiple choice benchmarks is loss-based -- the model is provided the answers choices concatenated with the prompt, and a question is scored as correct if the model assigns the highest likelihood to the ground truth answer (Brown 2020). This is fundamentally different than the generative, conversational evaluation setup for AlpacaEval and we believe that the benefits provided by NEFTune are better realized under those conditions.
> > >
> > > Finally, we do provide extensive analysis in Section 5.3 that the model's response is not just longer, but more comprehensive. That a more comprehensive answer has the potential ocassionally wrong in its details is a limitation that we point out, but we highlight that this does not imply that the model on average generates increased hallucinations.
> > >
> > > We thank you again for your continued engagement and we hope this clarifies a few final questions.

---

### Official Review · Reviewer_h2qY · 2023-11-02

**Soundness:** 3 good
**Presentation:** 3 good
**Contribution:** 3 good
**Rating:** 6
**Confidence:** 4

**Summary:**

This paper introduces NEFTune, a simple yet effective augmentation technique that improves the finetuning process of language models by adding noise to the embedding vectors during training. This is an empirical paper. The authors demonstrate that this method can substantially improve model performance, showcasing a huge increase from 29.79% to 64.69% on the AlpacaEval task when applied to the LLaMA-2-7B model fine tuned with Alpaca. Furthermore, the paper highlights NEFTune's ability to outperform strong baselines on more instruction datasets, reporting a 10% improvement for models trained with Evol-Instruct, an 8% improvement with ShareGPT, and an 8% improvement with OpenPlatypus. The authors also emphasize that even powerful models that are trained with RLHF, such as LLaMA-2-Chat, can still reap benefits from additional training with NEFTune. Overall, the paper establishes NEFTune as a valuable tool for enhancing the performance of language models across various tasks and datasets.

**Strengths:**

1) The research highlights a critical need to shift focus in the field of Large Language Models (LLMs) from predominantly concentrating on expanding datasets and model sizes, to also giving due attention to optimizing algorithms and regularization techniques. This is essential for enhancing model generalization and addressing overfitting, particularly when working with smaller instruction datasets. I completely agree with the authors on this. More emphasis must be laid on the regularization based methods for LLMs especially while finetuning on small datasets.
2) The authors introduce a simple yet effective regularization method of introducing noise into the embedding vector. Their comprehensive experimentation across a variety of instruct finetuning datasets and LLMs comparisons substantiates the effectiveness of their approach.
3) The authors deduce that an LLM trained with NEFTune yields longer and more comprehensive text generations, supported by qualitative examples where the NEFTune-generated text appears more detailed than its counterpart. However, they raise concerns about potential repetitiveness in these extended outputs. To address this, they employ n-gram and log diversity metrics, ultimately concluding that the text encompasses more information than mere repetition. They also conduct experiments to ascertain whether the generation of longer, comprehensive responses inherently leads to improved model performance. Their findings confirm that NEFTune outperforms this baseline, demonstrating its efficacy.
4) The authors attribute the substantial performance of NEFTune, surpassing that of baseline methods, to its ability to mitigate overfitting and enhance generalization. They support this claim with experimental results presented in Figures 4 and 5, although I do have some questions regarding this aspect.

**Weaknesses:**

I hope my comments will further strengthen the work.

1) The authors assert that their method outperforms baseline approaches by reducing overfitting. In this context, I would appreciate the inclusion of standard deviation values for the results, calculated over five random seeds. A small standard deviation with NEFTune would indeed validate its efficacy. Conversely, a large standard deviation might indicate that the stochastic nature of the noise added to the embedding layer doesn’t genuinely contribute to a regularization effect. Inclusion of standard deviation is insightful when finetuning an LLM on small datasets with a goal to mitigate overfitting [1, 2].

2) I noticed that the BLEU and Rouge-L scores on the training data for models with NEFTune applied appear to be relatively low. This observation leads me to wonder if the model is able to effectively learn from the training dataset. To gain a more comprehensive understanding of the training process, I would kindly request the inclusion of the training and validation loss curve plotted as a function of iterations for the +NEFT and without NEFT method. This additional information would be greatly appreciated and beneficial for a more thorough evaluation of the model's learning dynamics.

3) I've noticed that in the tables, such as Table 3, terms like "+NEFT 5" and similar are mentioned. It would be helpful if these terms could be explained or described directly in the captions of the tables to provide immediate context and clarity for readers.

4) Though adversarial ML literature (Zhu et al., 2019; Kong et al., 2022) has been cited for the choice of the noise α/√Ld. However, since it is the backbone for the entire work, I would suggest the authors to have a detailed discussion about it in the work explaining the reason behind this choice.

[1] Zhang, Haojie, et al. "Fine-Tuning Pre-Trained Language Models Effectively by Optimizing Subnetworks Adaptively." Advances in Neural Information Processing Systems 35 (2022): 21442-21454.

[2] Somayajula, Sai Ashish, et al. "Bi-level Finetuning with Task-dependent Similarity Structure for Low-resource Training." Findings of the Association for Computational Linguistics: ACL 2023. 2023.

**Questions:**

1) “While longer generations do score better, we see that no generation-time strategy comes close to the performance of NEFTune models.” The authors mention this in the paper, however it would be great if they can clarify further about this experiment and display those results in the paper.

**Details Of Ethics Concerns:**

.

---

> ### Author Response · Authors · 2023-11-16
> **Response to Reviewer h2qY**
>
> Thank you for your time, Reviewer h2qY.
>
> > While longer generations do score better, we see that no generation-time strategy comes close to the performance of NEFTune models.” The authors mention this in the paper, however it would be great if they can clarify further about this experiment and display those results in the paper.
>
> The sentence refers to the augmentation of the system prompt or changing the generation stragety to force longer outputs. These results can be found in Table 5.
>
> > Std Value of the Results
>
> Although we did not run different seeds for the finetuning, we can see from Table 6 that performance of difference noise values for the Alpaca dataset were equivalent. We are running different seeds and will add the results to the discussion if the experiments finish in time. For AlpacaEval, the reported Std Error is less than 2%.
>
> > Clarify "+NEFT 5"
>
> This refers to the alpha hyperparameter of NEFTune. We will make this clear in the updated verison of the paper.
>
> > Learning the Training Dataset
>
> From the training curves, we believe the model learns from training dataset. The training curves with noise is very similar to the training loss curve without noise. This suggests that the model is learning the task where the NEFTune models learns the original task plus the added noise, essentially learning a slightly different problem. Futhermore, we see that we recover some the training data when adding noise to the embedding when measuring with ROUGE-L, which can be found in Figure 6 in the updated Appendix.
>
> Please let us know if you have any more questions.

---

> ### Comment · Reviewer_h2qY · 2023-11-21
>
> 1) " generation-time strategy " This term is usually used for strategy such as beam search decoding, top-k sampling etc. So the result in table-5 does not support the claim.
>
> 2) "Std values" I am not really convinced by the result in table 6 because of the following reason. By changing the noise statistics the results still hold, which is counter intuitive since there has to be an inflection point beyond which adding noise should hurt the performance. There has to be a detailed study of how and why this algorithm is working. I understand the experiments are running for standard deviation however for this settings of the paper reporting the standard deviation is important. The setting assumed is low resource setting for LLMs and standard deviation defines the reliability of the algorithm proposed. I strongly encourage the authors to report them. "For AlpacaEval, the reported Std Error is less than 2%." this is a vague statement and I would want to see all the numbers.
>
> 3) As I mentioned above in the weakness, the authors did not clearly explain why their algorithm works. For instance, in the work stated above [2], they propose to represent a word embedding vector as a linear combination of all the related word embedding vectors weighted by the entries of a similarity matrix. The authors compare the proposed method with a baseline, BFTSS top-k random S, where they add random noise to the embedding representation of a word that improves robustness. This baseline outperforms the vanilla method however lags behind their method. I see NEFTune to be on the same lines as this work. However in their work, adding random noise outperforms vanilla in only some settings. So as I mentioned above, I urge the authors to inspect what is the noise statistics that improves the performance, why is it contributing to the better performance and also when it will deteriorate the performance. Another suggestion is, in the current form of the work, the authors are just adding noise to the embedding representation. This is uncontrolled noise and the interpretability of the method is less. I suggest the authors to add the baseline where the embedding vector of a word is represented as a linear combination of all the related word embedding vectors weighted by the entries of a similarity matrix initialized using cosine similarity of some pre-trained model embedding matrix [2]. They do not need to learn for the task-dependent similarity matrix. I believe from table 6 if the method works with a range of noise initialization then the suggested baseline should perform at par or even better as observed in [2]. It also brings some interpretability to the table.
>
> [2] "Bi-level Finetuning with Task-dependent Similarity Structure for Low-resource Training." Findings of the Association for Computational Linguistics: ACL 2023. 2023.
>
> I appreciate the authors for their research and especially recognizing the need to regularize to boost the performance. However, the current form of the work, as I mentioned in my previous comments, does not provide any understanding of why NEFTune works. The approach seems to be inspired from the previous methods in the literature, though a detailed comparison with those methods is not offered.  The standard deviation is necessary for methods dealing with low-resource. However, the results look promising and the paper is more on the empirical end.
>
> Nevertheless, since all my comments are not adequately answered, I would like to reduce my score from 8->6. That said, I recommend the authors to revise the work thoroughly since I believe it has a good potential.

---

> > ### Author Response · Authors · 2023-11-22
> > **Response to Reviewer h2qy (Part 1/2)**
> >
> > **Regarding Comment 1**
> >
> > Thank you for the clarifications. Regarding Comment 1, we humbly apologize for any confusion. We show the impact of different _prompting_ strategies in Table 5 as alternatives to NEFTune, which we referred to as "generation-time strategies", however, we agree that this specific verbiage was overly general. We apologize about the confusion in naming the prompting strategies "generation time strategies". We have rephrased that sentence to "no prompting strategy that we tested came close[...]".
> >
> > We did explore several decoding strategies common to LLaMA-type chat models. Results using top-p sampling and different temperature scales are shown in Table 7, and we see that the improvements NEFTune are relatively robust across generation strategies.
> >
> > **Regarding Statistical Analysis**
> >
> > We're further sorry if we've given the wrong impression in our response regarding statistical confidence in our results. In our previous response, we pointed to Table 6 as a placeholder, while we are running additional experiments to provide statistical estimates for every score result. Note that for a model trained with the Alpaca dataset, we now have collected results over five different seeds over one of the $\alpha=5$ values. We scored these runs on AlpacaEval with ChatGPT Evaluator scores, and, at the moment, for five seeds we have $60.50$, $60.00$, $58.32$, $59.07$, and $59.35$. With an overall mean result of $59.86$ Win Rate and standard error of $0.34$, we are now able to provide a better estimate of the performance of NEFTune across experiments. We are continuously running additional experiments with the aim of providing standard deviations for all experiments in our submission, but this will require additional time, due to the computational effort required.
> >
> > >  By changing the noise statistics the results still hold, which is counterintuitive since there has to be an inflection point beyond which adding noise should hurt the performance.
> >
> > Concerning the ablation of different noise levels, while we had originally run NEFTune with additional noise scales, i.e. values of alpha, we did not include those results in the initial submission copy. We have now re-run these experiments for the alpaca dataset, incorporated them into the updated manuscript, and we presented them in the table below for clarity. At low noise levels like $0.1$ and $1$ NEFTune does not meaningfully alter performance. On the other hand, slightly larger intermediate noise levels such $\alpha=50$ lead to reduced performance compared to the optimal settings of $\alpha=10$. For significantly larger noise values, such as $\alpha=100$, performance _does_ noticeably degrade as the generations become repetitive and nonsensical. Please see the table below for LLaMA-1 trained on the Alpaca dataset, which is ordered by AlpacaEval performance, where ChatGPT is the evaluator. We have now included this table with the full range of noise values as Figure 11 in the Appendix.
> >
> > | Alpha | AlpacaEval|
> > |:-----:| :--------:|
> > |  100  | 0.00      |
> > |  50   | 57.52     |
> > |  10   | 61.99     |
> > |   5   | 58.32     |
> > |   1   | 49.69     |
> > |  0.1  | 47.70     |
> > |   0   | 48.50     |
> >
> > This result shows that there is a minimum amount of noise required before NEFTune becomes effective and when there is too much noise NEFTune becomes less effective. The reason we suspect that NEFTune is no longer effective at this is that there are too many token flips occurring. Based on our extended analysis, see Figure 7 and 9 in the Appendix and Section A.3, we posit that NEFTune is effective when adding noise on the "intra-token" level. In Figure 7, we see that for the optimal level of noise, when the noisy embeddings are projected back to their nearest neighbor in the embedding matrix, no flips to neighboring tokens occur. In the range of alphas where we observe a significant fraction of flips to neighboring tokens, the performance of NEFTune is reduced.

---

> > > ### Author Response · Authors · 2023-11-22
> > > **Response to Reviewer h2qy (Part 2/2)**
> > >
> > > **Comparison with other augmentation strategies:**
> > >
> > > Although we were inspired by previous work, Table 9 shows that NEFTune worked better than a strong augmentation strategy for language modeling, FreeLB. In direct comparison, we find 67.45 for NEFTune vs 63.48 for FreeLB (AlpacaEval scores), on LLaMA-1 trained on Evol-Instruct data. Additionally, we also have compared to another popular augmentation technique at the token level, BPE-dropout, which provides no gains in chat model finetuning. Aside from this, baseline regularization strategies such as weight decay and dropout are standard in language model finetuning, but while these approaches are targets for hyperparameter tuning, they are known to have no drastic effects on chat model performance.
> > >
> > > |Setting| Alpaca | Alpaca+ BPE |  Alpaca+ FreeLB | Alpaca+ NEFTune|
> > > |-----| -------- | -------- | --------        |----------|
> > > |AlpacaEval (ChatGPT Evaluator)| 48.5     | 44.16     | 58.5     |    61.7    |
> > >
> > > **Why does NEFTune work?**
> > >
> > > We provide evidence that NEFTune behaves as a regularizer, and we provide an extensive analysis in Section 5.1 through 5.3 of the main work and Section A.3 of the appendix to support this characterization.
> > > We explain how the generations differ from those of the base model, a finding we also verify qualitatively and through a human study. Additionally, we show that the generations do not suffer from repetition and explore if there is a way through prompting that this difference can be explained.
> > >
> > > In particular, through the combination of ablation across noise scales in the response above (Table 6) and the analysis section on token flip rates at different noise scales, the updated version of this work begins to establish a more mechanistic understanding of how NEFTune works. Finally, the point you have raised about the relationship between our method and the technique proposed by [2] is a good question and would make for an insightful experiment.
> > >
> > > We apologize again for any confusion caused by our prior responses to your questions, and we hope we have helped clear up any remaining issues. In light of the additional experimental results collected and clarifications provided, we are wondering if you would consider raising your score?
> > >
> > > Thank you for your continued engagement and discussion.

---

> > > > ### Comment · Reviewer_h2qY · 2023-11-23
> > > >
> > > > Firstly, thank you for your response. It does clarify some of my questions.
> > > >
> > > > 1) I agree that running the experiments to get the standard deviation would take some time. I request the authors to include those numbers in the final revision. The provided empirical evidence for a model trained on Alpaca dataset under a noise statistics does offer a flavor however I believe we will have to observe the results on all the settings to comment on the robustness.
> > > >
> > > > 2) "We have now included this table with the full range of noise values as Figure 11 in the Appendix." It is insightful and does offer  more understanding of the method. Thank you!
> > > >
> > > > 3) I request the authors to add a paragraph in the methods section (sec 2) offering more intuitive explanation of what inspires them to choose the noise α/√Ld. Since it is the backbone for the entire work, I would suggest the authors to have a detailed discussion about it in the work explaining the reason behind this choice theoretically. I think including this will make it more understanding and interpretable.
> > > >
> > > >
> > > > However I have a few follow-up questions,
> > > >
> > > > 1) I agree with the provided explanation for the impact of noise in the method "In Figure 7, we see that for the optimal level of noise, when the noisy embeddings are projected back to their nearest neighbor in the embedding matrix, no flips to neighboring tokens occur. In the range of alphas where we observe a significant fraction of flips to neighboring tokens, the performance of NEFTune is reduced."
> > > >
> > > > However it makes me more convinced to see the performance of the method against the baseline in [2] atleast on one dataset and a model. No need to learn for a task specific similarity matrix. Just initialize the similarity matrix using the cosine similarity of a pretrained model embedding matrix. Because the embedding vector of a word is represented as a linear combination of all the related word embedding vectors weighted by the entries of a similarity matrix initialized using cosine similarity of some pre-trained model embedding matrix and going by the explanation provided there is an euclidean sphere of radius r (determined by the noise) around a particular word that when added to the embedding vector improves the robustness. I believe that such a euclidean sphere is better captured by the entries of a similarity matrix (in other words all related words) than a heuristically chosen noise.
> > > >
> > > > 2) "Aside from this, baseline regularization strategies such as weight decay and dropout are standard in language model finetuning, but while these approaches are targets for hyperparameter tuning, they are known to have no drastic effects on chat model performance." I think this statement is over generalizing, would be recommended to back it with references from literature. Also ** these approaches are targets for hyperparameter tuning**, I think the proposed method also has hyper parameters to tune such as $\alpha$.
> > > >
> > > > That said, I think this is a strong work but contingent upon inclusion of all the suggestions provided here and also by other reviewers. The major reason is that the interpretability of the proposed method is not clear. What are the reasons behind the choice (α/√Ld) and how it is achieving such huge gains. On the other hand, I am equally excited because the method is very simple to implement and requires no computational overhead and also provides good gains.

---

### Meta-Review · Area_Chair_sbfH · 2023-12-10

**Metareview:**

This paper shows that the performance of LLMs can be improved (sometimes substantially) by adding noise to embeddings.

While the authors do not give strong theoretical justifications for the observed gains, they validate them by thorough experiments.

The paper contributes a valuable simple approach and will likely spur follow-up works. I recommend acceptance as a poster.

During the rebuttal, the authors added more comparisons with other forms of regularization. These should go in the camera-ready version.

**Justification For Why Not Higher Score:**

- Lack of theoretical insights as to why the proposed approach works so well.
- Limited comparison with other forms of regularization

**Justification For Why Not Lower Score:**

Simple yet effective approach, thorough experiments

---

### Decision · Program_Chairs · 2024-01-16

Accept (poster)